# Ballistic macroscopic fluctuation theory via mapping to point particles

Jitendra Kethepalli ®,[1] Andrew Urilyon ®,[1] Tridib Sadhu ®,[2] Jacopo De Nardis ®,[1]

**1** Laboratoire de Physique Théorique et Modélisation, CNRS UMR 8089, CY Cergy Paris Universite, 95302 Cergy-Pontoise Cedex, France
**2** Department of Theoretical Physics, Tata Institute of Fundamental Research, Mumbai 400005, India

January 21, 2026

## Abstract

Ballistic Macroscopic Fluctuation Theory (BMFT) captures the evolution of fluctuations and correlations in systems where transport is strictly ballistic. We show that, for *generic integrable particle models*, BMFT can be constructed through a direct mapping onto ensembles of classical or quantum point particles. This mapping generalises the well-known correspondence between hard spheres and point particles: the two-body *scattering shift* now plays the role of an effective rod length for arbitrary interactions. Within this framework we re-derive both the full-counting statistics and the long-range correlation functions previously obtained by other means, thereby providing a unified derivation. Our results corroborate the general picture that all late-time fluctuations and correlations stem from the initial noise, subsequently convected by Euler-scale hydrodynamics.

# 1  Introduction and main results

Emergent dynamical properties in many-body systems pose a central challenge in physics due to the complex interplay between microscopic interactions and macroscopic behaviours. Hydrodynamics provides a powerful coarse-grained description: microscopic complexity is replaced by macroscopic equations of motion. By separating fast microscopic time scales from the slow evolution of conserved densities (and the modes coupled to them) and assuming local equilibrium, one derives hydrodynamic equations, typically partial differential equations, that govern the expectation values of those conserved-charge densities. Although conventional hydrodynamics can predict the time evolution of mean local observables, many other questions require information beyond simple averages, such as full-counting statistics of transported charge or multi-point connected correlation functions. The charge transport is computed by counting the amount of charge crossing a specific point, say the origin, in time $[0, T]$. For a system of particles with 1-particle phase space density at time $t$, it can be expressed as

$$\tilde{Q}_T = \int_{-\infty}^{\infty} d\theta \int_0^{\infty} dx \; h(\theta)\big\{\rho_T(x, \theta) - \rho_0(x, \theta)\big\}, \tag{1}$$

where $h(\theta)$ determines the nature of the transported charge. For instance, $h(\theta) = 1$ is for mass transport, $h(\theta) = \theta$ corresponds to momentum transport, etc. Similarly, the multi-point connected correlation functions, such as the 2-point function, given by

$$\mathcal{C}_{t_1, t_2}(x_1, \theta_1, x_2, \theta_2) = \big\langle \rho_{t_1}(x_1, \theta_1)\rho_{t_2}(x_2, \theta_2)\big\rangle^c, \tag{2}$$

rely on both thermodynamic fluctuations and hydrodynamic evolution. To capture such non-linear observables, one must restore these thermodynamic fluctuations and understand how they interact with the hydrodynamic evolution.

For diffusive systems, this program is realised by the *macroscopic fluctuation theory* (MFT) [1–5], which is valid for systems whose hydrodynamic current is proportional to the *gradient* of the charge density. Fluctuations are incorporated into the to hydrodynamics by (i) adding a stochastic noise term to the hydrodynamic current and (ii) sampling the initial hydrodynamic fields from a thermal fluctuating distribution. Exponentiating the noise via the Martin–Siggia–Rose-Janssen-De-Dominicis functional formalism and writing the corresponding initial action yields a path-integral description whose long-time behaviour can be evaluated by saddle-point methods [6–8]. MFT has been highly successful in characterising diffusive transport, generic correlations, and the full distribution of current fluctuations, see for example [9–23]. Ballistic systems, however, where the current is proportional to the charge density itself and not only to their derivatives, lie outside its scope.

This gap is filled by *ballistic macroscopic fluctuation theory* (BMFT) [24, 25]. The natural arena for BMFT is provided by generic integrable models, whose hydrodynamics

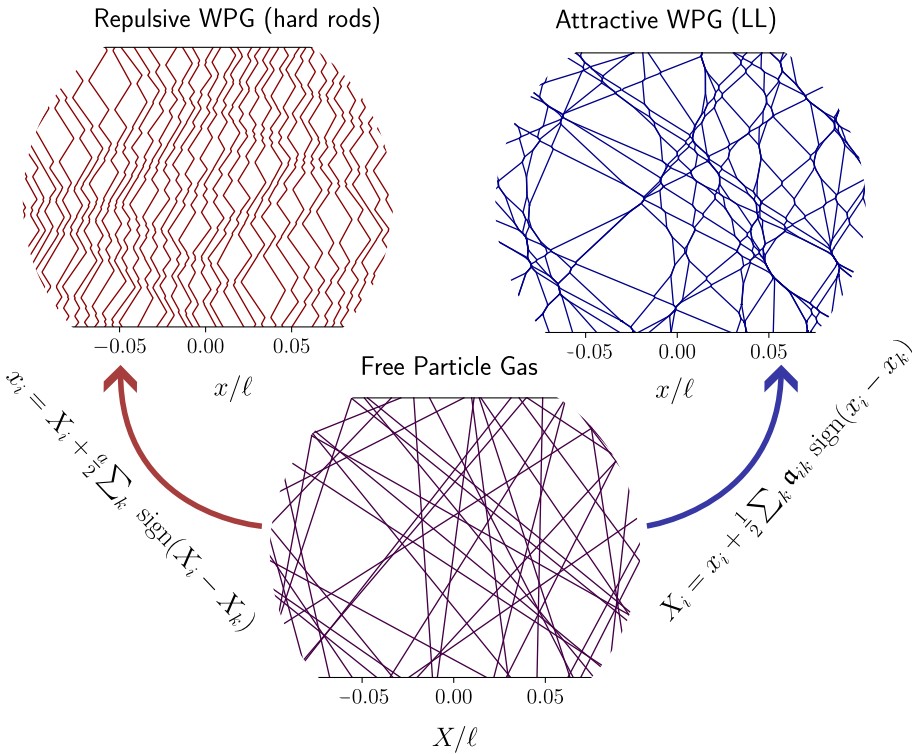

Figure 1: Cartoonish figure demonstrating the construction of the coordinates for a generic integrable model from the configuration of the free particle gas using the mapping in Eq. (5). Here we present two cases, (i) hard rods (red) with $\mathfrak{a}_{ij} = -a$ and (ii) Lieb-Liniger gas (blue) with $\mathfrak{a}_{ij} = 2/\big((\theta_i - \theta_j)^2 + 1\big)$. The mapping from the free particles trajectories $X_i(t)$ (evolving via $\dot{X}_i = \theta_i$) to interacting coordinates $x_i(t)$ requires inverting Eq. (5), which is a non-trivial task for generic $\mathfrak{a}_{i,j}$, and inversion is performed numerically. While the free particles have straight line trajectories, due to the mapping, the interacting coordinates acquire scattering shifts. Moreover, the distribution used for sampling the initial configurations determines the nature of the particles (i.e. Poissonian sampling for classical particles and Bernoulli sampling for Fermi-Dirac quantum particles).

is described by *generalised hydrodynamics* (GHD) [26, 27], see also [28–43, 43–57]. GHD involves an infinite family of conserved densities that can be interpreted as quasiparticles. Each quasiparticle moves ballistically, and two-body scattering merely renormalises its velocity; no bulk noise arises. The latter is indeed at the source of different anomalous fluctuations in integrable systems, see, for example, [58–63]. Therefore, BMFT requires only an accurate characterisation of the initial fluctuations, which then propagate ballistically under the non-linear Euler equations, producing non-trivial mesoscopic correlations. A transparent example is the one-dimensional gas of *hard rods* [33, 64–66] of length $a$. Because every collision is elastic and exchanges only momentum, the model is integrable. The position of the $i$-th rod, $x_i(t)$, can be mapped, at all times, to that of an auxiliary free particle, $X_i(t)$, through

$$x_i(t) = X_i(t) + \frac{a}{2} \sum_{k \neq i} \operatorname{sgn}(X_i(t) - X_k(t)) \tag{3}$$

with $X_i(t) = X_i(0) + \theta_i t$ at any time $t$, describing free particles (i.e. with $a = 0$) motion with velocities $\theta_i$. The initial velocities are therefore conserved, giving a set of $N$ conserved quantities of motion, and thus the hard rods represents the simplest integrable

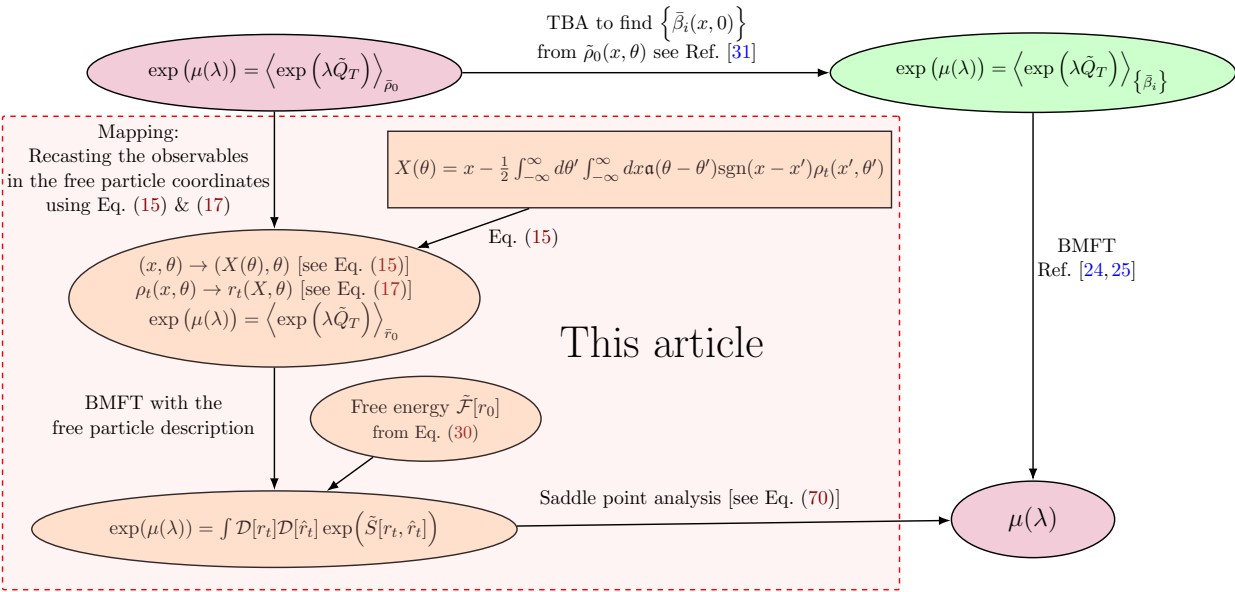

Figure 2: Schematic figure illustrating our modified BMFT approach for computing full counting statistics and correlation functions in integrable systems. In the standard BMFT framework introduced in Ref. [24, 25], the conserved charges are studied using the typical generalised temperatures $\{\bar{\beta}_i(x, 0)\}$ characterising the initial state. These temperatures can be obtained from the typical phase space density, $\bar{\rho}_0(x, \theta)$, using the thermodynamic Bethe ansatz (TBA) (see Ref. [31]). Here, we instead map the observable $\tilde{Q}_T$ [Eq. (1)] to the free particle coordinates via Eqs. (15) and (17). This allows us to apply the BMFT formalism, but now for the free particles and exploit its inherent simplicity. For example, the free energy of a free particle system is expressed in Eq. (30) and the system-specific details are hidden in $r_0(X, \theta) \leftarrow \rho_0(x, \theta)$.

model captured by GHD. In more general integrable models, the constant shift $a$ is replaced by a two-body scattering phase $\varphi(\theta, \theta')$ that depends on the incoming rapidities. Generic integrable systems can be viewed as deformations of the hard-rod gas in which the geometric shift $a$ is replaced by the model-dependent phase shift $\varphi$. In this article, we study the probability distribution of the net charge transport $\tilde{Q}_T$ [Eq. (1)] and the 2-point connected correlations [Eq. (2)] in a generic interacting integrable system. Our approach involves two main steps as described by the schematic flow diagram Fig. 2. First, we map the interacting system to a free particle system [via a generalisation of Eq. (3)] and recast these observables as functionals of the free particle coordinates. After the recasting, we use the BMFT formalism, but now for the free particles. Here, the typical equilibrium profile $\bar{r}_0(X, \theta)$ denotes the phase space density associated with the general Gibbs ensemble (GGE) used to prepare the system. It is obtained by fixing the generalized temperature $\{\bar{\beta}_i(x, 0)\}$ defining the GGE which determines the quasi-particle phase space density $\bar{\rho}_0(X, \theta)$ using the thermodynamic Bethe ansatz (TBA) (See Ref. [31]). This quasi particle density is then mapped to the free-particle coordinates using the generalised mapping in Eq. (17). This construction can be schematically as

$$\{\bar{\beta}_i(x)\} \xleftrightarrow{\text{TBA Ref. 32}} \bar{\rho}_0(x, \theta) \xleftrightarrow{\text{Mapping Eq.16}} \bar{r}_0(X, \theta). \tag{4}$$

## 1.1 GHD and its notation

Generic integrable systems are characterised by an extensive number of conserved modes, and therefore its large-scale hydrodynamics contains an infinite number of hydrodynamic density modes $q_i(x, t)$. On the other hand, as also pointed out in these past years by several recent works [44, 67–70], GHD can be reformulated (and explicitly derived from the semi-classical limit of Bethe wave functions) as a course grained dynamic of a gas of interacting wave packets (WPG), where a configuration of the particles' position and their moments $(x_i, \theta_i)_{i=1}^N$ can be mapped from that of the configurations of a free gas with coordinates $(X_i, \theta_i)_{i=1}^N$ by using the mapping, see also Fig. 1,

$$x_i(t) = X_i(t) - \frac{1}{2}\sum_{k\neq i} \mathfrak{a}_{ik}\, \mathrm{sgn}(x_i(t) - x_k(t)), \qquad \dot{X}_i(t) = v^{\mathrm{bare}}(\theta_i), \tag{5}$$

with $\mathfrak{a}_{ij}$ a generic (two-body) scattering amplitude. This mapping is merely a generalisation of the hard rods mapping of Eq. (3), where the effective rod length is replaced by the scattering shift as

$$a \to -\mathfrak{a}(\theta_i - \theta_j). \tag{6}$$

Replacing the hard rods length by the effective rod lengths for generic integrable models has so far been established only for the classical Toda model in Ref. [70]. For quantum integrable models, a heuristic derivation has been provided in Ref. [44]. More generally, it can be shown that any system of particles with coordinates $x_i$, such that at any time $t$ they can be mapped to the coordinates of free evolving particles $X_i$ [1] have a large-scale description given by the GHD equation. This procedure is: first introduce the 1-particle phase space density of quasiparticles, see also [69],

$$\rho_t(x, \theta) = \sum_{i=1}^N \delta(x - x_i)\delta(\theta - \theta_i). \tag{7}$$

and derive its equation of motion. Using Eq. (5) we obtain the evolution equation for $\rho_t(x, \theta)$ as [see Appendix A]

$$\partial_t \rho_t(x, \theta) + \partial_x\big(\rho_t(x, \theta)v^{\mathrm{eff}}_{[\rho_t(x,\cdot)]}(\theta)\big) = 0. \tag{8}$$

This equation is exact, and the effective velocity $v^{\mathrm{eff}}[\rho_t](x, \theta)$ is given in terms of the integral equation

$$v^{\mathrm{eff}}_{[\rho_t(x_i,\cdot)]}(\theta_i) = v^{\mathrm{bare}}(\theta_i) - \int \mathrm{d}\theta' \mathfrak{a}(\theta - \theta')\rho_t(x_i, \theta')(v^{\mathrm{eff}}_{[\rho_t(x_i,\cdot)]}(\theta_i) - v^{\mathrm{eff}}_{[\rho_t(x_i,\cdot)]}(\theta')). \tag{9}$$

Here $v^{\mathrm{bare}}(\theta)$ is the bare velocity of the quasiparticles (which in the case of the hard rods is given simply by $v^{\mathrm{bare}}(\theta) = \theta$). In the following, we shall focus on Galilean invariant models, where $v^{\mathrm{bare}}(\theta) = \theta$ and with bare momentum $k(\theta) = \theta$, and generalisations to a generic case are immediate.

The effective rod length $\mathfrak{a}_{ij}$ is generally given by the scattering shift of the underlying microscopic model. For example, for a Bethe ansatz integrable model, this is given by [36, 44, 67–69]

$$\mathfrak{a}_{ij} \equiv \mathfrak{a}(\theta_i - \theta_j) = \frac{2\pi\varphi(\theta_i - \theta_j)}{k'(\theta_i)}. \tag{10}$$

---

[1]Such a dynamical condition could be considered an alternative definition of integrability.

where $\varphi(\theta - \theta')$ denotes the derivative (divided by $2\pi$) of the scattering phase shift between particles of velocities (rapidities) $\theta$ and $\theta'$. Finally, it is convenient to express Eq. (8) in the normal modes density

$$n_t(x, \theta) = \frac{2\pi \rho_t(x, \theta)}{(1)^{\text{dr}}[n_t](x, \theta)}, \quad \text{with} \tag{11}$$

$$(1)^{\text{dr}}[n_t](x, \theta) \equiv (1)^{\text{dr}}_t(x, \theta) = 1 + \int_{-\infty}^{\infty} d\theta' \varphi(\theta - \theta') n_t(x, \theta')(1)^{\text{dr}}_t(x, \theta').$$

Here the notation $(h)^{\text{dr}}[\rho_t](x, \theta)$ denote the dressing of a generic function $h(\theta)$, which is also given in terms of an integral equation

$$(h)^{\text{dr}}[n_t](x, \theta) \equiv (h)^{\text{dr}}_t(x, \theta) = h(\theta) + \int_{-\infty}^{\infty} d\theta' \varphi(\theta - \theta') n_t(x, \theta')(h)^{\text{dr}}_t(x, \theta'). \tag{12}$$

The normal mode density effectively diagonalises the flux Jacobian in Eq. (8), which then factorises as a convective equation for each mode

$$\partial_t n_t(x, \theta) + v^{\text{eff}}_t(x, \theta)\partial_x n_t(x, \theta) = 0. \tag{13}$$

While GHD at the Euler scale captures ballistic transport of conserved quantities in integrable systems, higher-order corrections, such as the diffusive term, are important for their relaxation. GHD with diffusive contributions, often called the Navier-Stokes GHD [29,71], can be derived using the cumulant expansion as described in Ref. [72,73]. In this approach, the average current is expressed as a function of the average density and its higher-order correlations. For integrable models, these correlations arise from the fluctuations in the initial state, which are deterministically propagated via Euler GHD Eq. (13) with a non-linear effective velocity. This non-linearity is necessary, as integrable systems do not possess conventional scattering mechanisms. Instead, diffusion emerges from the coupling of different modes (quasi-particles) via the non-linear dependence of the effective velocity on the density of particles [see Eq. (9)]. In this work, we also study such correlations necessary for computing the diffusive correction to Euler GHD.

## 1.2 Interacting wave packet gas and mapping to point particles

The central idea of this paper is that all hydrodynamic fluctuations in interacting integrable models can be obtained by mapping their dynamics to that of point particles. In the phase space $(x, \theta)$, the position of the particles located in the region $[x - \Delta x/2, x + \Delta x/2] \cup [\theta - \Delta\theta/2, \theta + \Delta\theta/2]$ centered around $(x, \theta)$ can be mapped to that of the free particles with rapidity $\theta$, using Eq. (5) as

$$X(\theta) = x + \frac{1}{2} \sum_{x' \neq x} \sum_{\theta' \neq \theta} \Delta x \Delta\theta \rho_t(x', \theta')\mathfrak{a}(\theta - \theta')\text{sgn}(x - x'), \tag{14}$$

Here we note that the sum over the dummy index $j$ in Eq. (5) can be expressed as a sum over the coarse-grained $x'$ and $\theta'$ with the additional weight due to the particles at the coarse-grained location $x'$ and $\theta'$ i.e., $\sum_j = \sum_{x'} \sum_{\theta'} \Delta x \Delta\theta \rho_t(x', \theta')$, where $\rho_t(x, \theta)$ is the phase space density of the interacting particles at $(x, \theta)$. In the large-$N$ limit, the Eq. (14) becomes

$$X(\theta) = x + \frac{1}{2} \int_{-\infty}^{\infty} d\theta' \int_{-\infty}^{\infty} dx' \rho_t(x', \theta')\mathfrak{a}(\theta - \theta')\text{sgn}(x - x'). \tag{15}$$

Consider free coordinates $(X, k)$, where $k$ is the momenta which is parameterised by rapidity $\theta$, hereafter referred to as bare coordinates, and contrast these with the interacting coordinates $(x, \theta)$. By imposing this mapping and requiring that the number of particles be conserved $\int dX \frac{dk}{2\pi} r_t(X, k) = N = \int dx d\theta \, \rho_t(x, \theta)$ it follows that

$$r_t(X(\theta), k(\theta)) \frac{(k'(\theta))^{\mathrm{dr}}(x, \theta)}{2\pi} = \rho_t(x, \theta) \tag{16}$$

Here $\left(k'(\theta)\right)_t^{\mathrm{dr}}(x, \theta)$ is the Jacobian of the transformation $(x, \theta) \to (X(\theta), k(\theta))$. Henceforth, bare particle densities $r_t(X, \theta)$ are to be understood as being evaluated at the location $(X, k(\theta))$. In the case of a Galilean invariant model, $k(\theta) = \theta$, this yields a particularly convenient relation between the bare and interacting densities

$$r_t(X(\theta), \theta) = n_t(x, \theta). \tag{17}$$

Using Eq. (17), we can re-express the mapping in Eq. (15) as

$$x = X(\theta) - \frac{1}{2} \int_{-\infty}^{\infty} d\theta' \int_{-\infty}^{\infty} dX' \, r_t(X', \theta') \mathfrak{a}(\theta - \theta') \mathrm{sgn}(X(\theta') - X'). \tag{18}$$

Note that $X(\theta)$ depends on $r_t(X, \theta)$, $x$ and $\theta$ i.e., $X(\theta) \equiv X[r_t](x, \theta)$.

The hydrodynamic equation for the free particle density $r_t(X, \theta)$ can be obtained using Eq. (17) and Eq. (13), which gives

$$\partial_t r_t(X, \theta) + v_t(\theta) \partial_X r_t(X, \theta) = 0 \quad \text{with} \tag{19}$$

$$v_t(\theta) = \frac{(\theta)_t^{\mathrm{dr}}(x \to -\infty, \theta) + (\theta)_t^{\mathrm{dr}}(x \to \infty, \theta)}{2}. \tag{20}$$

Here, the phase-space density $r_t(X, \theta)$ propagates with velocity $v_t(\theta)$, itself depending on the density at the boundaries. When the boundaries are time independent $n_t(x \to \pm\infty, \theta) = n_\pm(\theta)$ and carry no net current i.e., satisfies $n_\pm(-\theta) = n_\pm(\theta)$ or when it vanishes ($n_\pm(\theta) = 0$) the propagation velocity becomes that of the bare velocity $v_t(\theta) = \theta$. For such boundaries, Eq. (19) describes the free point particles. In the rest of the article, we assume that the density at the boundaries ($x \to \pm\infty$) is not driven and set to zero, which makes $v_t(\theta) = \theta$. Using the Galilean invariance of Eq. (19), we can express its solution as

$$r_t(X, \theta) = r_0\big(X - t\theta, \theta\big). \tag{21}$$

Applying the mapping in Eq. (18), we conjecture that, by analysing the mesoscopic observables of the point particle system via the BMFT framework, one can infer the statistical behaviour of the appropriately modified observables in interacting integrable models.

## 1.3 Main results and organisation of the paper

The main result of this article is the reformulation of BMFT for integrable systems through mapping [see Eq. (5)] their configuration to that of the point particles. Using our approach, we study the following two quantities

1. The full counting statistics corresponding to $\tilde{Q}_T$ defined in Eq. (1) by computing the generating function [see Eq. (45) for (non-interacting) point particles system and Eq. (78) for interacting system]

$$\text{FCS:} \quad \mu(\lambda) = \log \left[ \Big\langle \exp\Big(\lambda \tilde{Q}_T\Big) \Big\rangle_{\bar{r}_0} \right], \tag{22}$$

where the average is over fluctuations in the initial state characterised by average phase space density $\bar{r}_0(Z, \theta)$ in the point particle coordinates.

2. The 2-point normal mode density correlation [see Eq. (108)]

$$\text{correlations:} \quad \langle n_{t_1}(x_1,\theta_1) n_{t_2}(x_2,\theta_2)\rangle^c. \tag{23}$$

is obtained by studying the generating function

$$\text{Generating function :} \quad \Big\langle \exp\Big(T\lambda_1 n_{t_1}\big(x_1,\theta_1\big) + T\lambda_2 n_{t_2}\big(x_2,\theta_2\big)\Big)\Big\rangle_{\bar{n}_0}. \tag{24}$$

This paper is organised as follows: in section 2 we introduce the BMFT for point particles, showing how it can also be done for generic particle statistics, and in particular quantum statistics. In section 3 we use the BMFT for generic classical and quantum interacting systems to compute the full counting statistics. In section 4, we study the 2-point normal mode correlations for these systems. Finally in section 5 we conclude with an outlook and potential directions for this approach.

## 2  BMFT for point particles

A useful quantity for studying transport is the net charge transferred across the origin during a finite but long time $T$ as defined in Eq. (1). As this quantity depends on the initial configuration, it is a random variable whose statistics are known as the full counting statistics (FCS). Although computing FCS can be challenging in general, the formalism of MFT for diffusive systems and BMFT for ballistic systems offers a systematic framework. To illustrate BMFT, consider the simplest example, free point particles. At Euler scale their phase space density $r_t(X,\theta)$ obeys

$$\partial_t r_t(X,\theta) + \theta \partial_X r_t(X,\theta) = 0. \tag{25}$$

The net charge transfer over a time interval $T$ from left to right of the origin can be expressed as [using Eq. (1)]

$$\tilde{Q}_T = \int_{-\infty}^{\infty} d\theta \int_0^{\infty} dX \ h(\theta)\Big(r_T(X,\theta) - r_0(X,\theta)\Big), \tag{26}$$

where the function $h(\theta)$ is an arbitrary function. This is exactly the difference between the total charge on the right of the origin at time $T$ and at time 0. Here, to obtain Eq. (26) from Eq. (1) we used $r_t(X,\theta) = \rho_t(x,\theta)$ and $X = x$, which can be easily obtained by setting $\mathfrak{a}(\theta - \theta') = 0$ in Eqs. (15) and (17).

Clearly, the observable $\tilde{Q}_T$ is deterministic for any given initial configuration $r_0(X,\theta)$ once it evolves via Eq. (25). Its randomness originates solely from the stochastic nature of the initial configuration $r_0(X,\theta)$, which is determined by the preparation of our system. A natural choice is when all the particles are independently sampled to have a

$$\text{typical equilibrium profile :} \quad \bar{r}_0(X,\theta), \tag{27}$$

so that the realized profile $r_0(X,\theta)$ fluctuates around $\bar{r}_0(X,\theta)$. The probability of observing the initial phase-space density is then given by the large deviation principle with the free energy cost of creating $r_0(X,\theta)$ from $\bar{r}_0(X,\theta)$, see Fig. 3 and Appendix B for the explicit calculations. The probability density functional denoted by $\mathcal{P}_0[r_0(X,\theta)]$ is given by

$$\mathcal{P}_0\left[r_0(X,\theta)\right] \asymp \exp\Big(-\tilde{\mathcal{F}}[r_0(X,\theta)]\Big), \quad \text{where} \tag{28}$$

$$\tilde{\mathcal{F}}[r_0(X,\theta)] = \int_{-\infty}^{\infty} d\theta \int_{-\infty}^{\infty} dY \ \big[f(r_0) - f(\bar{r}_0) - (r_0 - \bar{r}_0)f'(\bar{r}_0)\big].$$

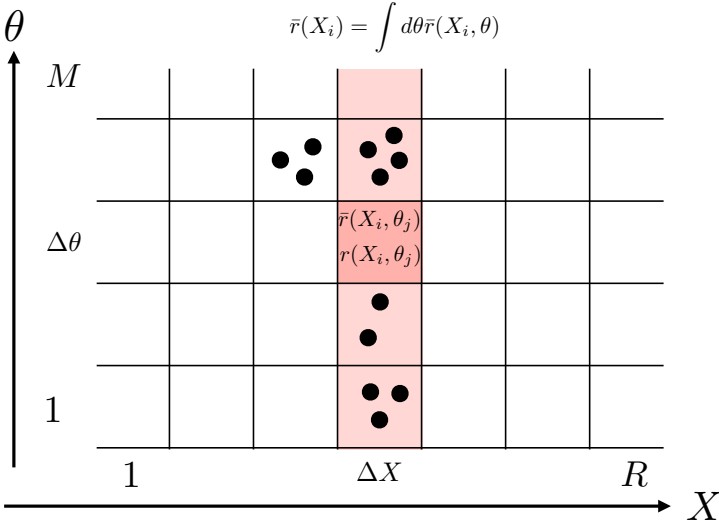

Figure 3: Schematic plot of the $(X, \theta)$ phase space, partitioned into $R$ strips of size $\Delta X$ which are labeled by index $i \in \{1, 2, ..., R\}$ and centered around position $X_i$ as shown by the pink strip. Each of these strips are further partitioned into $M$ smaller subsystems of size $\Delta \theta$ which are labeled by index $(., j)$ and have particles with velocities between $[\theta_j - \Delta\theta/2, \theta_j + \Delta\theta/2]$. The subsystem labeled by $(i, j)$ centered around $(X_i, \theta_j)$ has $n_{i,j} \equiv r(X_i, \theta_j)\Delta X \Delta \theta$ number of particles and its typical value is $\bar{n}_{i,j} \equiv \bar{r}(X_i, \theta_j)\Delta X \Delta \theta$.

Here, the free energy per unit phase-space volume is given by

$$f(r_0) = \begin{cases} r_0 \log r_0 - r_0 & \text{for Classical particles,} \\ r_0 \log r_0 + \eta\big(1 - \eta r_0\big) \log\big(1 - \eta r_0\big) & \text{for Quantum particles} \end{cases} \tag{29}$$

where $\eta = \pm 1$ are for Fermions and Bosons, respectively. Using the Eq. (29) in Eq. (28) we obtain the free energy cost as

$$\tilde{\mathcal{F}}[r_0] = \int_{-\infty}^{\infty} d\theta \int_{-\infty}^{\infty} dX \; G(r_0, \bar{r}_0) \;\; \text{with} \tag{30}$$

$$G(r_0, \bar{r}_0) = \begin{cases} r_0 \log\left(\frac{r_0}{\bar{r}_0}\right) - \big(r_0 - \bar{r}_0\big) & \text{for Classical particles,} \\ r_0 \log\left(\frac{r_0}{\bar{r}_0}\right) + (1 - r_0) \log\left(\frac{1-r_0}{1-\bar{r}_0}\right) & \text{for Fermions,} \\ r_0 \log\left(\frac{r_0}{\bar{r}_0}\right) - (1 + r_0) \log\left(\frac{1+r_0}{1+\bar{r}_0}\right) & \text{for Bosons.} \end{cases}$$

For the case of radiative modes and solitons

The statistical properties of $\tilde{Q}_T$ can be understood by computing the generating function

$$\left\langle \exp\left(\lambda \tilde{Q}_T\right) \right\rangle_{\bar{r}_0} = \int \mathcal{D}\left[r_t(X, \theta)\right] \mathcal{P}_t[r_t(X, \theta)] \exp\left(\lambda \tilde{Q}_T\right), \tag{31}$$

where the angular brackets $\langle \, * \, \rangle_{\bar{r}_0}$ with the subscript $\bar{r}_0$ represents an average over the initial density profile. Here, the probability density function of any profile $r_t(x, \theta)$ can be obtained using the propagator that evolves the initial density profile, which is given by

$$\mathcal{P}_t[r_t(X, \theta)] = \delta\left[\partial_t r_t(X, \theta) + \partial_X \theta r_t(X, \theta)\right] \exp\left(-\tilde{\mathcal{F}}[r_0]\right). \tag{32}$$

In Eq. (32), the delta functional ensures that the phase space density satisfies the hydrodynamic evolution equation Eq. (25). Substituting Eq. (32) into the expression of generating function in Eq. (31), we get

$$
\left\langle \exp\left(\lambda\tilde{Q}_T\right)\right\rangle_{\bar{r}_0} = \int \mathcal{D}\left[r_t(X,\theta)\right] \delta\left[\partial_t r_t(X,\theta) + \partial_X \theta r_t(X,\theta)\right] \exp\left(-\tilde{\mathcal{F}}[r_0]\right) \exp\left(\lambda\tilde{Q}_T\right).
\tag{33}
$$

Using the Fourier representation for the delta-functional Eq. (33) can be expressed as

$$
\left\langle \exp\left(\lambda\tilde{Q}_T\right)\right\rangle_{\bar{r}_0} = \int \mathcal{D}\left[r_t(X,\theta)\right] \mathcal{D}\left[\hat{r}_t(X,\theta)\right] \exp\left(\tilde{\mathcal{S}}[r_t(X,\theta),\hat{r}_t(X,\theta)]\right),
\tag{34}
$$

where $\hat{r}_t(X,\theta)$ is the field conjugate to $r_t(X,\theta)$ and the action $\tilde{\mathcal{S}}[r_t(X,\theta),\hat{r}_t(X,\theta)]$ is given by

$$
\tilde{\mathcal{S}}[r_t(X,\theta),\hat{r}_t(X,\theta)] = -\int_{-\infty}^{\infty} d\theta \int_{-\infty}^{\infty} dX \int_0^T dt\, \hat{r}_t(X,\theta)\left[\partial_t r_t(X,\theta) + \partial_X \theta r_t(X,\theta)\right]
$$
$$
- \tilde{\mathcal{F}}[r_0(X,\theta)] + \lambda\tilde{Q}_T, \quad \text{with } \tilde{Q}_T \text{ given in Eq. (26).}
\tag{35}
$$

To investigate ballistic transport the position and time is rescaled as $X = T\,Z$ and $t = T\,\tau$, with $Z$ and $\tau$ being the rescaled position and time variables. In these rescaled variables, the action is

$$
\tilde{\mathcal{S}}[r_t(X,\theta),\hat{r}_t(X,\theta)] = T\mathcal{S}[q_\tau(Z,\theta),p_\tau(Z,\theta)] \quad \text{where}
$$
$$
\mathcal{S}[q_\tau(Z,\theta),p_\tau(Z,\theta)] = -\int_{-\infty}^{\infty} d\theta \int_{-\infty}^{\infty} dZ \int_0^1 d\tau\, p_\tau(Z,\theta)\left[\partial_\tau q_\tau(Z,\theta) + \partial_Z \theta q_\tau(Z,\theta)\right]
$$
$$
- \mathcal{F}[q_0(Z,\theta)] + \lambda Q_1,
\tag{36}
$$

where $q_\tau(Z,\theta)$ and $p_\tau(Z,\theta)$ are functions of the rescaled variables

$$
q_\tau(Z,\theta) = r_t(X,\theta), \quad \bar{q}_\tau(Z,\theta) = \bar{r}_t(X,\theta), \quad p_\tau(Z,\theta) = \hat{r}_t(X,\theta)
\tag{37}
$$
$$
\text{with} \quad \tau = \frac{t}{T} \quad \text{and} \quad Z = \frac{X}{T}.
\tag{38}
$$

In Eq. (36), the scaled charge and the scaled free energy cost are given, respectively, by

$$
Q_1 = \frac{1}{T}\tilde{Q}_T = \int_{-\infty}^{\infty} d\theta \int_0^{\infty} dZ\, h(\theta)\left(q_1(Z,\theta) - q_0(Z,\theta)\right),
\tag{39}
$$
$$
\mathcal{F}[q_0(Z,\theta)] = \frac{1}{T}\tilde{\mathcal{F}}[r_0(X,\theta)] = \int_{-\infty}^{\infty} d\theta \int_{-\infty}^{\infty} dZ\, G\left(q_0(Z,\theta),\bar{q}_0(Z,\theta)\right),
\tag{40}
$$

with $G(q_0,\bar{q}_0)$ given in Eq. (30). As the action in Eq. (36) scales linearly with time $T$, we use the saddle point calculation to compute the generating function in Eq. (34). We obtain the following saddle point equations

$$
\frac{\delta\mathcal{S}[q_\tau,p_\tau]}{\delta p_\tau(Z,\theta)} = \partial_\tau q_\tau^*(Z,\theta) + \partial_Z \theta q_\tau^*(Z,\theta) = 0,
\tag{41a}
$$
$$
\frac{\delta\mathcal{S}[q_\tau,p_\tau]}{\delta q_\tau(Z,\theta)} = \partial_\tau p_\tau^*(Z,\theta) + \partial_Z \theta p_\tau^*(Z,\theta) = 0,
\tag{41b}
$$
$$
\frac{\delta\mathcal{S}[q_\tau,p_\tau]}{\delta q_1(Z,\theta)} = p_1^*(Z,\theta) - \lambda\Theta(z)h(\theta) = 0,
\tag{41c}
$$
$$
\frac{\delta\mathcal{S}[q_\tau,p_\tau]}{\delta q_0(Z,\theta)} = -p_0^*(Z,\theta) + \lambda\Theta(Z)h(\theta) - \frac{\delta\mathcal{F}[q_0]}{\delta q_0(Z,\theta)} = 0,
\tag{41d}
$$

where $\cdot^*$ denotes the saddle point value. Eqs. (41a) and (41b) are obtained from the variation of action Eq. (36) due to the auxiliary field $p_\tau(Z,\theta)$ and $q_\tau(Z,\theta)$, respectively. While Eq. (41c) and Eq. (41d) are obtained by variation due to the densities at time-boundaries *i.e.*, $q_1(Z,\theta)$ and $q_0(Z,\theta)$, respectively. Since Eq. (41a) and Eq. (41b) are Galilean invariant, they can be solved as

$$q_\tau^*(Z,\theta) = q_0^*(Z - \tau\theta, \theta) \ \text{ and } \ p_\tau^*(Z,\theta) = p_0^*(Z - \tau\theta, \theta). \tag{42}$$

Using Eq. (42) in Eqs. (41d) and (41c) and substituting the free energy from Eq. (40), we can obtain the saddle point initial density, $q_0^*(Z,\theta)$, by solving

$$\frac{q_0^*(Z,\theta)}{1 - \eta q_0^*(Z,\theta)} = \frac{\bar{q}_0(Z,\theta)}{1 - \eta \bar{q}_0(Z,\theta)} \exp\left(\lambda h(\theta)\{\Theta(Z+\theta) - \Theta(Z)\}\right), \tag{43}$$

where $\Theta(Z)$ is Heavy-side step function and setting $\eta = 0$ we get the saddle point solution for classical systems and $\eta = \pm 1$ for Bosons and Fermions, respectively. The scaled saddle point action is obtained by substituting Eq. (42) in Eq. (36), which gives

$$\mathcal{S}[q_\tau^*(Z,\theta), p_\tau^*(Z,\theta)] = -\mathcal{F}[q_0^*(Z,\theta)] + \lambda Q_1^*, \tag{44}$$

where $Q_1^*$ is the scaled charge evaluated by substituting $q_\tau(Z,\theta) = q_\tau^*(Z,\theta)$ in the expression of $Q_1$, Eq. (39), which gives $Q_1 = Q_1^*$. The cumulant generating function $\mu(\lambda)$ is then obtained by substituting the saddle point initial condition $(q_0^*(Z,\theta))$ in Eq. (34) to get

$$\exp\left(\mu(\lambda)\right) = \left\langle \exp\left(\lambda\tilde{Q}_T\right) \right\rangle_{\bar{r}_0} \asymp \exp\left(T\mathcal{S}[q_\tau^*(Z,\theta), p_\tau^*(Z,\theta)]\right),$$
$$\mu(\lambda) \asymp T\left(\lambda Q_1^* - \mathcal{F}[q_0^*(Z,\theta)]\right). \tag{45}$$

By simplifying Eq. (45) for the classical systems, we get

$$\mu(\lambda) = T \int_0^\infty d\theta \int_{-\theta}^0 dZ\, \bar{q}_0(Z,\theta)\left[\exp(\lambda h(\theta)) - 1\right]$$
$$+ T \int_{-\infty}^0 d\theta \int_0^{-\theta} dZ\, \bar{q}_0(Z,\theta)\left[\exp(-\lambda h(\theta)) - 1\right]. \tag{46}$$

While for the Quantum system, it is given by

$$\mu(\lambda) = T\eta \int_0^\infty d\theta \int_{-\theta}^0 dZ \log\left(1 + \eta\bar{q}_0(Z,\theta)\left[\exp\left(\lambda h(\theta)\right) - 1\right]\right) \tag{47}$$
$$+ T\eta \int_{-\infty}^0 d\theta \int_0^{-\theta} dZ \log\left(1 + \eta\bar{q}_0(Z,\theta)\left[\exp\left(-\lambda h(\theta)\right) - 1\right]\right).$$

Here, we recall that $\eta = \pm 1$ is for Fermions and Bosons, respectively. Also recall that the typical profile $\bar{q}_0(Z,\theta) = \bar{r}_0(TZ,\theta)$ as given in Eq. (37). We can compute all the cumulants by taking derivatives of $\mu(\lambda)$, and find that as $\mu(\lambda) \propto T$ Eq. (45), all the cumulants must grow linearly with $T$. We note that for the classical systems with $h(\theta) = 1$ *i.e.* for mass transport, the expression of $n^{\text{th}}$ cumulant is much simple and given by

$$\kappa_n = T\left(Q_+ + (-1)^n Q_-\right), \tag{48}$$

where

$$Q_- = \int_{-\infty}^0 d\theta \int_0^{-\theta} dZ\, \bar{q}_0(Z,\theta), \quad \text{and} \quad Q_+ = \int_0^\infty d\theta \int_{-\theta}^0 dZ\, \bar{q}_0(Z,\theta). \tag{49}$$

A derivation based on the microscopic approach also gives the same result as shown in the Appendix C.

Using the Legendre duality between the cumulant generating function $\mu(\lambda)$ and the rate function $\mathcal{I}(Q_1)$, we can compute the expression of the rate function corresponding to the probability distribution of $Q_1$ as

$$\mathcal{P}(Q_1) \asymp \exp\big(-T\mathcal{I}(Q_1)\big) \ \ \text{with} \ \ \mathcal{I}(Q_1) = \lambda^* Q_1 - \frac{\mu(\lambda^*)}{T}, \tag{50}$$

where $\lambda^*$ is obtained by solving

$$\frac{d}{d\lambda}\mu(\lambda)\Big|_{\lambda=\lambda^*} = TQ_1 \ \ \text{for any } Q_1. \tag{51}$$

The expression of the rate function in Eq. (50) can be simplified by substituting the expression of the $\mu(\lambda)$ given in Eq. (44) to get

$$\mathcal{I}(Q_1) = \mathcal{F}[q_0^*(Z,\theta)], \tag{52}$$

where $q_0^*(Z,\theta)$ is obtained from solving Eq. (43) and setting $\lambda = \lambda^*$ with $\lambda^*$ obtained from Eq. (51).

## 2.1 FCS in partitioning protocol

The partitioning protocol constitutes an ideal setting to study transport in ballistic systems, see for example [26, 27, 74]. This protocol involves initialising the system into two semi-infinite domains characterised by a homogeneous particle density $r_\pm(\theta)$, which are joined at the origin. Inside the light-cone generated by the dynamics, the phase-space density is given by (see also [26, 27, 36, 74]),

$$\bar{r}_0(X,\theta) = \bar{r}_+(\theta)\Theta(X) + \bar{r}_-(\theta)\Theta(-X), \tag{53}$$

where $\Theta(X)$ is Heaviside step function. To understand its evolution consider the dynamics along a ray $\xi = X/t$. With these ray coordinates eq. (25) can be expressed as

$$(\xi-\theta)\partial_\xi \tilde{r}(\xi,\theta) = 0 \ \ \text{where} \ \ \tilde{r}(\xi,\theta) = r_t(X,\theta). \tag{54}$$

So starting from the initial condition Eq. (53), the free point particle density evolves as

$$\tilde{r}(\xi,\theta) = \bar{r}_-(\theta)\Theta(\theta-\xi) + \bar{r}_+(\theta)\Theta(\xi-\theta). \tag{55}$$

In this setup, we can simplify the expressions of the cumulant generating function given in Eq. (46) and Eq. (47) to get

$$\mu(\lambda) = T\int_{-\infty}^{\infty} d\theta \ |\theta|\tilde{r}_0(\theta)\Big[\exp\big(\lambda h(\theta)\mathrm{sgn}(\theta)\big) - 1\Big], \ \ \text{for Classical}, \tag{56}$$

$$\mu(\lambda) = \eta T\int_{-\infty}^{\infty} d\theta \ |\theta|\log\Big(1 + \eta\tilde{r}_0(\theta)\Big[\exp\big(\lambda h(\theta)\mathrm{sgn}(\theta)\big) - 1\Big]\Big), \ \ \text{for Quantum}, \tag{57}$$

where we defined $\tilde{r}_0(\theta) = \tilde{r}(\xi = 0,\theta)$ and it is given by Eq. (55). Using Eq. (56) and (57), we can compute the cumulants by taking their derivative with $\lambda$. The first four cumulants

are then readily computed as

$$\kappa_1 = T \int_{-\infty}^{\infty} d\theta \; \theta \; h(\theta) \tilde{r}_0(\theta), \tag{58a}$$

$$\kappa_2 = T \int_{-\infty}^{\infty} d\theta \; |\theta| \big(h(\theta)\big)^2 \tilde{r}_0(\theta) \big(1 - \eta \tilde{r}_0(\theta)\big), \tag{58b}$$

$$\kappa_3 = T \int_{-\infty}^{\infty} d\theta \; \theta \big(h(\theta)\big)^3 \tilde{r}_0(\theta) \big(1 - \eta \tilde{r}_0(\theta)\big) \big(1 - 2\eta \tilde{r}_0(\theta)\big), \tag{58c}$$

$$\kappa_4 = T \int_{-\infty}^{\infty} d\theta \; |\theta| \big(h(\theta)\big)^4 \tilde{r}_0(\theta) \big(1 - \eta \tilde{r}_0(\theta)\big) \big(1 - 6\eta \tilde{r}_0(\theta) + 6\eta^2 \tilde{r}_0(\theta)^2\big). \tag{58d}$$

# 3 BMFT for interacting systems via mapping to point particles

Equipped with the BMFT description of point particles, we proceed to the case of interacting integrable models. Our approach involves working with the free particle coordinates and making use of the transformation (15) to comment on the interacting case. In this section, we study the full counting statistics for general integrable models by studying the statistical properties of $\tilde{Q}_T$ defined in Eq. (1), which can be represented in the free coordinates as

$$\tilde{Q}_T = \int_{-\infty}^{\infty} d\theta \int_{L_T(\theta)}^{\infty} dX \; h(\theta) r_T(X, \theta) - \int_{-\infty}^{\infty} d\theta \int_{L_0(\theta)}^{\infty} dX \; h(\theta) r_0(X, \theta), \tag{59}$$

where we have used Eq. (17) and the variable transformation Eq. (15) in Eq. (1). The location of the origin in the free coordinates at any time $t$ is denoted by $L_t(\theta)$ and obtained by setting $x = 0$ in Eq. (15), which gives

$$L_t(\theta) = \frac{1}{2} \int_{-\infty}^{\infty} d\theta' \int_{-\infty}^{\infty} dX \; r_t(X, \theta') \mathfrak{a}(\theta - \theta') \mathrm{sgn}\big(L_t(\theta') - X\big). \tag{60}$$

Although the observable in Eq. (59) is deterministic and solely determined by the initial condition, $r_0(X, \theta)$, the fluctuations in the initial configuration cause it to behave like a random variable. For studying the statistical properties of $\tilde{Q}_T$, we use the approach described in Section 2 and compute

$$\big\langle \exp\big(\lambda \tilde{Q}_T\big) \big\rangle_{\bar{r}_0} = \int \mathcal{D}[r_t(X, \theta)] \mathcal{D}[\hat{r}_t(X, \theta)] \exp\left( \tilde{S}\big[r_t(X, \theta), \hat{r}_t(X, \theta)\big] \right), \tag{61}$$

where the action $\tilde{S}\big[r_t(X, \theta), \hat{r}_t(X, \theta)\big]$ is given by

$$\tilde{S}\big[r_t(X, \theta), \hat{r}_t(X, \theta)\big] = - \int_{-\infty}^{\infty} d\theta \int_{-\infty}^{\infty} dX \int_0^T dt \; \hat{r}_t(X, \theta) \big[\partial_t r_t(X, \theta) + \theta \partial_X r_t(X, \theta)\big]$$

$$- \tilde{\mathcal{F}}[r_0(X, \theta)] + \lambda \tilde{Q}_T, \quad \text{with} \quad \tilde{Q}_T \text{ given in Eq. (59).} \tag{62}$$

Here, we have assumed that the statistical nature of the initial profile, when described in the point particle density $r_0(x, \theta)$, is governed by the probability density functional with the large deviation function $\tilde{\mathcal{F}}[r_0]$ given in Eq. (28) along with Eq. (30). While a general proof is not available, we present a combinatorial calculation in the Appendix B.2 to demonstrate that the free energy of the hard rods when expressed in terms of the point particles is indeed described by Eq. (28) with the free energy per unit volume given

by Eq. (30). This is a well motivated assumption, as the Hamiltonian governing the mapped point particles is non-interacting (see Ref. [67]) and the equilibrium properties are determined by the free energy of the non-interacting system. Hence, one expects that the large deviation functional for the initial state coincides with the free energy of the non-interacting system.

Since we are concerned with the ballistic scaling, we choose $X = TZ$ and $t = T\tau$ where $Z$ and $\tau$ are the rescaled position and time. In the rescaled variables, the action can be expressed as

$$\tilde{\mathcal{S}}\big[r_t(X,\theta), \hat{r}_t(X,\theta)\big] = T\mathcal{S}\big[q_\tau(Z,\theta), p_\tau(Z,\theta)\big], \quad \text{where} \tag{63}$$

$$\mathcal{S}\big[q_t(Z,\theta), p_t(Z,\theta)\big] = -\int_{-\infty}^{\infty} d\theta \int_{-\infty}^{\infty} dZ \int_0^T dt \; p_\tau(Z,\theta)\left[\partial_\tau q_\tau(Z,\theta) + \theta\partial_z q_\tau(Z,\theta)\right]$$
$$- \mathcal{F}[q_0(Z,\theta)] + \lambda Q_1. \tag{64}$$

Here $\mathcal{F}[q_0(Z,\theta)]$ is rescaled free energy cost given in Eq. (40), the scaled charge $Q_1$ is given by

$$Q_1 = \frac{1}{T}\tilde{Q}_T = \int_{-\infty}^{\infty} d\theta \int_{l_1(\theta)}^{\infty} dz \; h(\theta)q_1(Z,\theta) - \lambda\int_{-\infty}^{\infty} d\theta \int_{l_0(\theta)}^{\infty} dz \; h(\theta)q_0(Z,\theta) \tag{65}$$

and $q_\tau(Z,\theta), p_\tau(Z,\theta)$ are the phase-space density and its conjugate in the rescaled variables

$$q_\tau(Z,\theta) = r_t(X,\theta) \quad \text{with} \quad \tau = \frac{t}{T} \quad \text{and} \quad z = \frac{X}{T}, \tag{66}$$

$$\bar{q}_\tau(Z,\theta) = \bar{r}_\tau(X,\theta) \tag{67}$$

$$p_\tau(Z,\theta) = \hat{r}_t(X,\theta). \tag{68}$$

The lengths $l_0(\theta)$ and $l_1(\theta)$ in Eq. (65) are given by

$$l_\tau(\theta) = \frac{L_t(\theta)}{T} \quad \text{with} \quad t = \tau T \quad \text{and} \tag{69}$$

$$l_\tau(\theta) \equiv l_\tau[q_\tau(Z,\theta)] = \frac{1}{2}\int_{-\infty}^{\infty} d\theta' \int_{-\infty}^{\infty} dz' q_\tau(z',\theta)\mathfrak{a}(\theta - \theta')\text{sgn}\big(l_\tau(\theta') - z'\big).$$

As the action in Eq. (63) scales linearly with time $T$, for large $T$, we use the saddle point calculation to compute the generating function in Eq. (61). By setting the variation of $\mathcal{S}[q_\tau, p_\tau]$ at $q_\tau^*(Z,\theta)$ and $p_\tau^*(Z,\theta)$ to zero the following saddle point equations are obtained

$$\frac{\delta\mathcal{S}[q_\tau, p_\tau]}{\delta p_\tau(Z,\theta)} = \partial_\tau q_\tau^*(Z,\theta) + \theta\partial_z q_\tau^*(Z,\theta) = 0, \tag{70a}$$

$$\frac{\delta\mathcal{S}[q_\tau, p_\tau]}{\delta q_\tau(Z,\theta)} = \partial_\tau p_\tau^*(Z,\theta) + \theta\partial_z p_\tau^*(Z,\theta) = 0, \tag{70b}$$

$$\frac{\delta\mathcal{S}[q_\tau, p_\tau]}{\delta q_1(Z,\theta)} = \frac{\lambda}{2}\Big(h(\theta) - \text{sgn}\big(l_1^*(\theta) - z\big)c_1^*(\theta)\Big) - p_1^*(Z,\theta) = 0, \tag{70c}$$

$$\frac{\delta\mathcal{S}[q_\tau, p_\tau]}{\delta q_0(Z,\theta)} = p_0^*(Z,\theta) - \frac{\lambda}{2}\Big(h(\theta) - \text{sgn}\big(l_0^*(\theta) - z\big)c_0^*(\theta)\Big) - \frac{\delta\mathcal{F}[q_0^*(Z,\theta)]}{\delta q_0(Z,\theta)} = 0, \tag{70d}$$

where $c_0^*(\theta)$ and $c_1^*(\theta)$ are

$$c_0^*[q_0^*](x = 0,\theta) \equiv c_0^*(\theta) = (h)_0^{\text{dr}}(x = 0,\theta) \quad \text{dressing with} \quad n_0^*(0,\theta) = q_0^*(l_0^*(\theta),\theta) \tag{71}$$

$$c_1^*[q_1^*](x = 0,\theta) \equiv c_1^*(\theta) = (h)_T^{\text{dr}}(x = 0,\theta) \quad \text{dressing with} \quad n_T^*(0,\theta) = q_1^*(l_1^*(\theta),\theta). \tag{72}$$

To obtain these saddle point equations, we also assumed that the density at the boundary is fixed and vanishes. The main task is now to solve the saddle point equation Eqs. (70a)-(70d). Since Eq. (70a) and Eq. (70b) are Galilean invariant Euler equations of a free system, the solution is given by

$$q_\tau^*(Z,\theta) = q_0^*(z - \tau\theta, \theta), \qquad p_\tau^*(Z,\theta) = p_0^*(z - \tau\theta, \theta). \tag{73}$$

In Eq. (70d), we use Eq. (70c) which is back propagated using Eq. (73) and substitute the expression of the scaled free energy, Eq. (40), to find

$$\frac{q_0^*(Z,\theta)}{1 - \eta q_0^*(Z,\theta)} = \frac{\bar{q}_0(Z,\theta)}{1 - \eta \bar{q}_0(Z,\theta)} \exp\left(\frac{\lambda}{2}\left[c_0^*(\theta)\mathrm{sgn}(l_0^*(\theta) - z) - c_1^*(\theta)\mathrm{sgn}(l_1^*(\theta) - \theta - z)\right]\right), \tag{74}$$

where $l_0^*(\theta)$, $l_1^*(\theta)$, $c_0^*(\theta)$ and $c_1^*(\theta)$ are given by

$$l_0^*(\theta) = \frac{1}{2}\int_{-\infty}^\infty d\theta' \int_{-\infty}^\infty dz' q_0^*(z',\theta')\mathfrak{a}(\theta - \theta')\mathrm{sgn}(l_0(\theta') - z'), \tag{75a}$$

$$l_1^*(\theta) = \frac{1}{2}\int_{-\infty}^\infty d\theta' \int_{-\infty}^\infty dz' q_1^*(z',\theta')\mathfrak{a}(\theta - \theta')\mathrm{sgn}(l_1(\theta') - z'), \tag{75b}$$

$$c_0^*(\theta) = (h)_0^{\mathrm{dr}}(x = 0, \theta) \quad \text{dressing with} \quad n_0^*(0,\theta) = q_0^*(l_0^*(\theta), \theta),$$
$$c_0^*(\theta) = h(\theta) + \int_{-\infty}^\infty d\theta' \varphi(\theta - \theta')q_0^*(l_0^*(\theta'), \theta')c_0^*(\theta'), \tag{75c}$$

$$c_1^*(\theta) = (h)_T^{\mathrm{dr}}(x = 0, \theta) \quad \text{dressing with} \quad n_T^*(0,\theta) = q_1^*(l_1^*(\theta), \theta),$$
$$c_1^*(\theta) = h(\theta) + \int_{-\infty}^\infty d\theta' \varphi(\theta - \theta')q_1^*(l_1^*(\theta'), \theta')c_1^*(\theta'). \tag{75d}$$

Note that all the functions are implicitly dependent on $\lambda$. Substituting the saddle point density profile, obtained by solving Eq. (74) and Eq. (75), in Eq. (64) we find that the saddle point action is given by

$$\mathcal{S}\left[q_t^*(Z,\theta), p_t^*(Z,\theta)\right] = -\mathcal{F}[q_0^*(Z,\theta)] + \lambda Q_1^*. \tag{76}$$

where the scaled charge at the saddle point density is obtained by substituting $q_0^*(Z,\theta)$ obtained from solving Eq. (74) in Eq. (65), which gives

$$Q_1^* = \int_{-\infty}^\infty d\theta \int_{l_1^*(\theta)}^\infty dz\, h(\theta)q_1^*(Z,\theta) - \int_{-\infty}^\infty d\theta \int_{l_0^*(\theta)}^\infty dz\, h(\theta)q_0^*(Z,\theta) \tag{77}$$

We can now compute the cumulant generating function given in Eq. (61) by using Eq. (63) and Eq. (76) which gives

$$\boxed{\mu(\lambda) = T\left(\lambda Q_1^* - \mathcal{F}[q_0^*(Z,\theta)]\right) \quad \text{where } q_0^*(Z,\theta) \text{ is obtained from Eqs. (74)}.} \tag{78}$$

Here recall that $\mathcal{F}[q]$ is the free energy cost given in Eq. (40). Using the Legendre duality, we can express the rate function as [see Eq. (50)]

$$\mathcal{I}(Q_1) = \mathcal{F}[q_0^*(Z,\theta)], \tag{79}$$

where $q_0^*(Z,\theta)$ is obtained from solving Eq. (74) at $\lambda = \lambda^*$ and $\lambda^*$ is obtained by solving

$$\left.\frac{d}{d\lambda}\mu(\lambda)\right|_{\lambda=\lambda^*} = TQ_1 \quad \text{for any } Q_1. \tag{80}$$

Consequently, we obtain the charge at the saddle point (governed by $\lambda$) using

$$\frac{d}{d\lambda}\mu(\lambda) = TQ_1^* \quad \text{where } Q_1^* \text{ is given in Eq. (77).} \tag{81}$$

Hence, the first cumulant is obtained by setting $\lambda = 0$ in Eq. (81) to get

$$\kappa_1 = T\int_{-\infty}^{\infty} d\theta \int_{\bar{l}_1(\theta)}^{\infty} dZ \; h(\theta)\bar{q}_1(Z,\theta) - T\int_{-\infty}^{\infty} d\theta \int_{\bar{l}_0(\theta)}^{\infty} dZ \; h(\theta)\bar{q}_0(Z,\theta). \tag{82}$$

Here $\bar{l}_\tau(\theta)$ is obtained from Eq. (69) by replacing $q_\tau(Z,\theta)$ with $\bar{q}_\tau(Z,\theta)$. To compute the second cumulant, we take a derivative with respect to $\lambda$ on both sides of Eq. (81) and then set $\lambda = 0$ to get

$$\kappa_2 = \left[\frac{d^2}{d\lambda^2}\mu(\lambda)\right]_{\lambda=0} \tag{83}$$

$$= T\frac{1}{4}\int_{-\infty}^{\infty} d\theta \int_{-\infty}^{\infty} dZ \; \left(\bar{c}_0(\theta) - \bar{c}_1(\theta)\right)^2 \bar{q}_1(Z,\theta)\left(1 - \eta\bar{q}_1(Z,\theta)\right) \tag{84}$$

$$+ T\int_{\theta_c}^{\infty} d\theta \int_{\bar{l}_1(\theta)}^{\bar{l}_0(\theta)+\theta} dZ \; \bar{c}_0(\theta)\bar{c}_1(\theta)\bar{q}_1(Z,\theta)\left(1 - \eta\bar{q}_1(Z,\theta)\right)$$

$$+ T\int_{-\infty}^{\theta_c} d\theta \int_{\bar{l}_0(\theta)+\theta}^{\bar{l}_1(\theta)} dZ \; \bar{c}_0(\theta)\bar{c}_1(\theta)\bar{q}_1(Z,\theta)\left(1 - \eta\bar{q}_1(Z,\theta)\right),$$

where $\theta_c$ solves $\theta_c = \bar{l}_1(\theta_c) - \bar{l}_0(\theta_c)$ and $\bar{c}_{0/1}(\theta)$ are given in Eq. (75c) and Eq. (75d) with $q_{0/1}^*(Z,\theta)$ replaced by $\bar{q}_{0/1}(Z,\theta)$. Calculating the cumulant generating function, or equivalently rate function for arbitrary typical profiles, becomes increasingly complex. We therefore focus on two cases: (i) homogeneous and (ii) partitioning protocol (inhomogeneous).

For homogeneous initial conditions, the phase space density is given by

$$\bar{\rho}_0(x,\theta) = \bar{\varrho}\,\bar{p}(\theta). \tag{85}$$

In the normal mode coordinates, we get

$$\bar{n}_0(x,\theta) = \frac{2\pi\bar{\rho}_0(x,\theta)}{(1)_0^{\mathrm{dr}}(x,\theta)} \quad \text{with} \quad (1)_0^{\mathrm{dr}}(x,\theta) = 1 + \varrho\,\langle\varphi(\theta-\theta')\rangle_{\bar{p}(\theta)}, \tag{86}$$

where $\langle*\rangle_{\bar{p}(\theta)}$ represents average with $\bar{p}(\theta)$. Expressing the density in the point particle coordinate, and after scaling, we get

$$\bar{q}_0(Z(\theta),\theta) = n_0(x,\theta) = \frac{2\pi\bar{\rho}_0(x,\theta)}{(1)_0^{\mathrm{dr}}(x,\theta)} \quad \text{with} \quad Z(\theta) = \frac{X(\theta)}{T}, \tag{87}$$

where $X(\theta)$ is given in Eq. (18). The rate function is obtained by substituting Eq. (73) with $\bar{q}_0(Z,\theta)$ from Eq. (87) in Eq. (79). In the Fig. 4, we show a plot of the rate function given in Eq. (79) for the hard rods that is obtained by solving Eq. (75) iteratively. We compare them with our numerical simulations. The plot shows that the distribution is non-Gaussian. We next compute the cumulants for the inhomogeneous profile with the partitioning protocol setup.

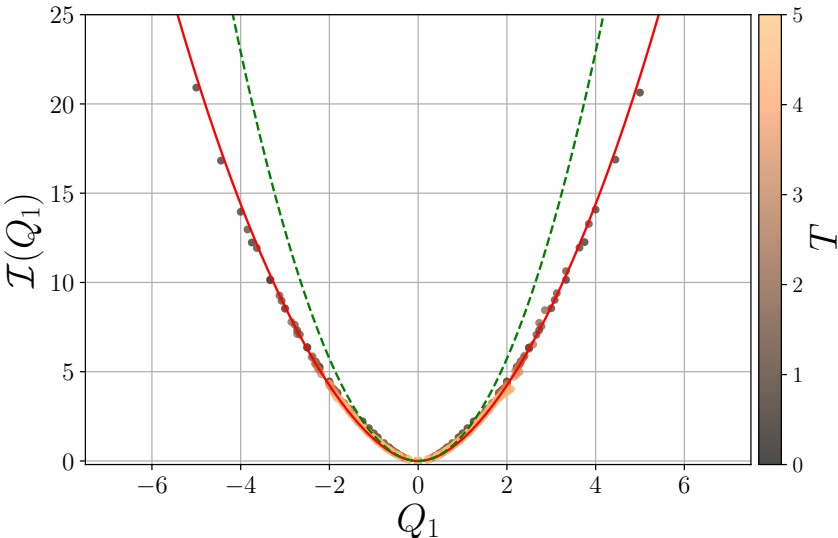

Figure 4: Plot showing the rate function [Eq. (79) with $\bar{q}_0(Z, \theta)$ given by Eq. (87)] for the density observable $Q_1 = \tilde{Q}_T/T$, with $h(\theta) = 1$, in a system of hard rods of length $a = \frac{1}{4}$ and mean number of particles $\langle N \rangle = 2^{14}$ for $T \in [0.5, 5]$. Initial momenta are sampled from a Gaussian distribution with zero mean and variance set by the temperature $1/\beta = 1$. These configurations are mapped to interacting coordinates using the mapping in Eq. (5). The point particles evolve freely ($\dot{X}_i = \theta_i$) and the interacting coordinates are computed at later times $T$ using the mapping Eq. (5). The observable $Q_1$ is evaluated from $10^8$ independent trajectories to construct the rate function. The typical part of the distribution aligns with the Gaussian distribution (green dashed line) with variance set by $\kappa_2$ [Eq. (83) with $\bar{q}_1(Z, \theta)$ from Eq. (87)]. However, the tails *i.e.*, the atypical fluctuations behave differently from Gaussian as shown by the theoretical rate function (red solid lines). These highlight the non-Gaussian nature of the FCS.

## 3.1   Charge fluctuations in the partitioning protocol

For the partitioning protocol, the phase space density is described by [26, 27, 36, 74]

$$\bar{\rho}_0(x, \theta) = \bar{\rho}_+(\theta)\Theta(x) + \bar{\rho}_-(\theta)\Theta(-x), \tag{88}$$

where $\Theta(x)$ is Heavy-side step function. In the normal mode coordinates [see Eq. (11)], we get

$$\bar{n}_0(x, \theta) = \frac{2\pi \bar{\rho}_0(x, \theta)}{(1)_0^{\mathrm{dr}}(x, \theta)} = \bar{n}_+(\theta)\Theta(x) + \bar{n}_-(\theta)\Theta(-x), \tag{89}$$

where the functions

$$\bar{n}_\pm(\theta) = \frac{2\pi \bar{\rho}_\pm(\theta)}{(1)_\pm^{\bar{\mathrm{dr}}}(\theta)} \quad \text{with} \quad (1)_\pm^{\bar{\mathrm{dr}}}(\theta) = 1 + \int_{-\infty}^{\infty} d\theta' \varphi(\theta - \theta') \bar{n}_\pm(\theta') (1)_\pm^{\bar{\mathrm{dr}}}(\theta'). \tag{90}$$

To study the charge fluctuations in the partitioning protocol, analysing the evolution of normal mode densities in the ray coordinates is convenient. In these coordinates, we track the evolution of the density along the ray $\xi = x/t$, and Eq. (13) in the ray coordinates becomes

$$\partial_\xi \tilde{n}(\xi, \theta)(\xi - \tilde{v}^{\mathrm{eff}}(\xi, \theta)) = 0 \quad \text{where} \quad \tilde{n}(\xi, \theta) = n_t(x, \theta). \tag{91}$$

So starting from the initial condition Eq. (89), the normal mode density evolves as

$$\tilde{n}(\xi, \theta) = n_-(\theta)\Theta\big(v^{\text{eff}}(\xi, \theta) - \xi\big) + n_+(\theta)\Theta\big(\xi - v^{\text{eff}}(\xi, \theta)\big) \quad \text{and equivalently as}$$

$$\tilde{n}(\xi, \theta) = n_-(\theta)\Theta\big(\theta - \theta^*(\xi)\big) + n_+(\theta)\Theta\big(\theta^*(\xi) - \theta\big), \quad \text{where} \quad v^{\text{eff}}(\xi, \theta^*) = \xi. \tag{92}$$

This suggests that the density along the ray $\xi = 0$ ($x = 0, t > 0$) does not evolve. Hence for this protocol, $\bar{c}_0(\theta) = \bar{c}_1(\theta) = \big(h\big)_t^{\text{dr}}(x = 0, \theta)$ [see Eqs. (75c) and (75d)].

We can express the phase space density in Eq. (89) in the point particle coordinates as

$$\bar{q}_0(Z, \theta) = \bar{n}_+(\theta)\mathbb{1}\big(Z > \bar{l}_0(\theta)\big) + \bar{n}_-(\theta)\mathbb{1}\big(Z < \bar{l}_0(\theta)\big). \tag{93}$$

The density at the point corresponding to the origin $x = 0$ evolves as $\bar{q}_\tau(\bar{l}_\tau(\theta), \theta) = \bar{q}_0(\bar{l}_\tau(\theta) - \tau\theta, \theta)$ and we obtain

$$\bar{q}_\tau(\bar{l}_\tau(\theta), \theta) = \bar{n}_+(\theta)\mathbb{1}\big(\bar{l}_\tau(\theta) > \bar{l}_0(\theta) + \tau\theta\big) + \bar{n}_-(\theta)\mathbb{1}\big(\bar{l}_\tau(\theta) < \bar{l}_0(\theta) + \tau\theta\big). \tag{94}$$

Here note that $(\bar{l}_\tau(\theta), \theta)$ is the location in the point particle's coordinates corresponding to the point $(x = 0, \theta)$.

For this protocol, we can compute the cumulants by taking the derivative of the cumulant generating $\mu(\lambda)$ function with $\lambda$ and setting $\lambda = 0$. Here we present the expression for the first three cumulants:

$$\kappa_1 = T \int_{-\infty}^{\infty} d\theta \big(\bar{l}_0(\theta) + \theta - \bar{l}_1(\theta)\big) \, h(\theta)\bar{q}_1(\bar{l}_1(\theta), \theta). \tag{95a}$$

$$\kappa_2 = T \int_{-\infty}^{\infty} d\theta \big|\bar{l}_1(\theta) - \bar{l}_0(\theta) - \theta\big| \, \big(\bar{c}_1(\theta)\big)^2 \bar{q}_1(\bar{l}_1(\theta), \theta)\big(1 - \eta\bar{q}_1(\bar{l}_1(\theta), \theta)\big). \tag{95b}$$

$$\kappa_3 = \left[\frac{d^3}{d\lambda_h^3}\mu_I^{(h)}(\lambda_h)\right]_{\lambda_h = 0}$$

$$= T \int_{-\infty}^{\infty} d\theta \big(\bar{c}_0(\theta)\big)^3\big(\theta + \bar{l}_0(\theta) - \bar{l}_1(\theta)\big)\bar{q}_1(\bar{l}_1(\theta), \theta)(1 - \eta\bar{q}_1(\bar{l}_1(\theta), \theta))(1 - 2\eta\bar{q}_1(\bar{l}_1(\theta), \theta))$$

$$+ T \int_{-\infty}^{\infty} d\theta \bar{q}_1(\bar{l}_1(\theta), \theta)(1 - \eta\bar{q}_1(\bar{l}_1(\theta), \theta))\bar{c}_0(\theta)\big|\theta + \bar{l}_0(\theta) - \bar{l}_1(\theta)\big| \times \tag{95c}$$

$$\times \int_{-\infty}^{\infty} d\psi \varphi_0^{\text{dr}}(\theta - \psi)\Big[3\big(\bar{c}_1(\psi)\big)^2\Big]\bar{q}_1(\bar{l}_1(\psi), \psi)(1 - \eta\bar{q}_1(\bar{l}_1(\psi), \psi))\text{sgn}\big(\psi + \bar{l}_0(\psi) - \bar{l}_1(\psi)\big)$$

Note that by setting $\varphi(\theta - \theta') = 0$ in Eq. (95), we obtain the result for the point particles given in Eq (58).

In the ray coordinates [Eq. (92)], the expression of the first three cumulants is given by

$$\kappa_1 = \int_{-\infty}^{\infty} d\theta \, v_{[\tilde{\rho}(0,.)]}^{\text{eff}}(\theta) \, h(\theta)\tilde{\rho}(0, \theta), \tag{96a}$$

$$\kappa_2 = \int_{-\infty}^{\infty} d\theta \, \big|v_{[\tilde{\rho}(0,.)]}^{\text{eff}}(\theta)\big| \, \big[\bar{c}_1(\theta)\big]^2 \tilde{\rho}(0, \theta)\big(1 - \eta\tilde{n}(0, \theta)\big), \tag{96b}$$

$$\kappa_3 = \int_{-\infty}^{\infty} d\theta \, v_{[\tilde{\rho}(0,.)]}^{\text{eff}}(\theta) \, \big(\bar{c}_1(\theta)\big)^3 \tilde{\rho}(0, \theta)(1 - \eta\tilde{n}(0, \theta))(1 - 2\eta\tilde{n}(0, \theta)) \tag{96c}$$

$$+ \int_{-\infty}^{\infty} d\theta\tilde{\rho}(0, \theta)(1 - \eta\tilde{n}(0, \theta))\bar{c}_1(\theta)\big|v_{[\tilde{\rho}(0,.)]}^{\text{eff}}(\theta)\big| \times$$

$$\times \int_{-\infty}^{\infty} d\psi \varphi_0^{\text{dr}}(\theta - \psi)\Big[3\big(\bar{c}_1(\psi)\big)^2\Big]\tilde{n}(0, \psi)(1 - \eta\tilde{n}(0, \psi))\text{sgn}\big(v_{[\tilde{\rho}(0,.)]}^{\text{eff}}(\psi)\big)$$

where the effective velocity $v^{\text{eff}}_{[\tilde{\rho}(0,.)]}(\theta)$ is given in Eq. (9), $\bar{c}_1(\theta) = \left(h\right)^{\text{dr}}(\xi = 0, \theta)$ and $\tilde{\rho}(0,\theta) = \tilde{n}(0,\theta)\left(1\right)^{\text{dr}}(\xi = 0, \theta)$. Here we also used the relation

$$(\theta)^{\text{dr}}(\xi = 0, \theta) = \theta + \bar{l}_0(\theta) - \bar{l}_1(\theta). \tag{97}$$

This can be obtained by computing $\bar{l}_0(\theta) + \theta - \bar{l}_1(\theta)$ and it satisfies

$$\bar{l}_1(\theta) - \bar{l}_0(\theta) - \theta = \int_{-\infty}^{\infty} d\theta' \left[\bar{l}_1(\theta') - \bar{l}_0(\theta') - \theta'\right]\tilde{n}(0,\theta')\mathfrak{a}(\theta - \theta') - \theta. \tag{98}$$

Using the dressing operation on Eq. (98), we obtain Eq. (97). The expression of the first three cumulants in Eq. (96) agrees with the result obtained in Ref. [24, 25]. Here we note that identifying the cumulants with derivatives of the scaled generating function in Eqs. (82), (83) and (96) requires sufficient regularity of the latter. In particular, if the finite-time generating functions are uniformly bounded in a complex neighborhood of the origin, Bryc's regularity condition applies [75]. This condition ensures analyticity of the scaled cumulant generating function and implies the validity of the central limit theorem as a consequence of finite scaled cumulants. When Bryc's condition is violated, non-analytic behavior may arise, and both the standard identification of cumulants via derivatives and the central limit theorem may fail. It is known to be violated in some integrable models see Ref. [61]. Therefore, our results should be understood as applying to regimes where the scaled cumulant generating function is regular.

## 4  Two-point normal modes correlations

In this section, we compute the correlation of the normal-mode phase-space density at $(x_1, \theta_1, t_1)$ with $(x_2, \theta_2, t_2)$ *i.e.*,

$$\text{Correlation}: \ \mathcal{C}_{t_1,t_2}(x_1, \theta_1, x_2, \theta_2) = \langle n_{t_1}(x_1, \theta_1)n_{t_2}(x_2, \theta_2)\rangle^c. \tag{99}$$

The correlation can be computed using BMFT formalism by studying the generating function

$$\exp\left(T\mathcal{G}(\lambda_1, \lambda_2)\right) = \left\langle \exp\left(T\lambda_1 n_{t_1}\left(x_1, \theta_1\right) + T\lambda_2 n_{t_2}\left(x_2, \theta_2\right)\right)\right\rangle_{\bar{n}_0}, \tag{100}$$

where $0 < t_1, t_2 < T$ and $\bar{n}_0(x,\theta)$ defines the typical initial state of the system. To compute Eq. (100) using our formalism, we first map the observable $n_{t_1}(x_1, \theta_1)$ and $n_{t_2}(x_2, \theta_2)$ to the bare coordinates using Eq. (17) to get

$$n_{t_1}(x_1, \theta_1) = r_{t_1}(\mathfrak{L}_1(\theta_1), \theta_1) \ \text{ and } \ n_{t_2}(x_2, \theta_2) = r_{t_2}(\mathfrak{L}_2(\theta_2), \theta_2). \tag{101}$$

Here, the $\mathfrak{L}_1(\theta_1) \equiv \mathfrak{L}[r_{t_1}](x_1, \theta)$ and $\mathfrak{L}_2(\theta_2) \equiv \mathfrak{L}[r_{t_2}](x_2, \theta)$ denote the position in the free particle coordinate corresponding to the point $(x_1, \theta_1, t_1)$ and $(x_2, \theta_2, t_2)$, respectively. They can be obtained from Eq. (18) as

$$\mathfrak{L}_1(\theta) = x_1 + \frac{1}{2}\int_{-\infty}^{\infty} d\theta' \int_{-\infty}^{\infty} dX' \ \mathfrak{a}(\theta - \theta')\text{sgn}\left(\mathfrak{L}_1(\theta') - X'\right)r_{t_1}(X', \theta'), \tag{102}$$

$$\mathfrak{L}_2(\theta) = x_2 + \frac{1}{2}\int_{-\infty}^{\infty} d\theta' \int_{-\infty}^{\infty} dX' \ \mathfrak{a}(\theta - \theta')\text{sgn}\left(\mathfrak{L}_2(\theta') - X'\right)r_{t_2}(X', \theta'). \tag{103}$$

The generating function in Eq. (100) can be reexpressed in terms of the free particle BMFT as

$$\exp\Big(T\mathcal{G}(\lambda_1,\lambda_2)\Big) = \Big\langle \exp\Big(T\lambda_1 r_{t_1}\big(\mathfrak{L}_1(\theta_1),\theta_1\big) + T\lambda_2 r_{t_2}\big(\mathfrak{L}_2(\theta_2),\theta_2\big)\Big)\Big\rangle_{\bar{r}_0}, \qquad (104)$$

where $\bar{r}_0(Z,\theta)$ is the typical phase-space density in the free coordinates. This is evaluated in Appendix D where we show the generating function is given by

$$\mathcal{G}(\lambda_1,\lambda_2) = \mathcal{S}\big[q_\tau^*,p_\tau^*\big] \quad \text{with} \qquad (105)$$
$$\mathcal{S}\big[q_\tau^*,p_\tau^*\big] = -\mathcal{F}[q_0^*] + \lambda_1 q_{\tau_1}^*\big(\mathfrak{l}_1^*(\theta_1),\theta_1\big) + \lambda_2 q_{\tau_2}^*\big(\mathfrak{l}_2^*(\theta_2),\theta_2\big).$$

Here $q_\tau^*(Z,\theta) = q_0^*(Z-\theta\tau,\theta)$ is given in Eq. (167) of Appendix D. To compute the 2-point normal-mode correlation, we use the Legendre duality $i.e.$,

$$\frac{d}{d\lambda_1}\mathcal{G}(\lambda_1,\lambda_2)\Big|_{\lambda_1=0} = q_{\tau_1}^*\big(\mathfrak{l}_1^*(\theta_1),\theta_1\big)\Big|_{\lambda_1=0}. \qquad (106)$$

Taking the derivative of Eq. (106) with respect to $\lambda_2$ and setting $\lambda_2=0$, we get the 2-point correlation as

$$\mathcal{C}_{t_1,t_2}(x_1,\theta_1,x_2,\theta_2) = \frac{d}{d\lambda_2}\Big[q_{\tau_1}^*\big(\mathfrak{l}_1^*(\theta_1),\theta_1\big)\Big|_{\lambda_1=0}\Big]\Big|_{\lambda_2=0}. \qquad (107)$$

Performing the derivative with $\lambda_2$ and setting $\lambda=0$ we obtain the correlation and by expressing them in the unscaled point particle density, we get

$$
\begin{aligned}
&\mathcal{C}_{t_1,t_2}(x_1,\theta_1,x_2,\theta_2) \hspace{6cm} (108)\\
&= \bar{r}_{t_1}(\bar{\mathfrak{L}}_1(\theta_1),\theta_1)\delta(\bar{\mathfrak{L}}_2(\theta_2)-\bar{\mathfrak{L}}_1(\theta_2)-(t_2-t_1)\theta_2)\delta(\theta_1-\theta_2)\\
&+ \bar{r}_{t_1}(\bar{\mathfrak{L}}_1(\theta_1),\theta_1)\big(\mathfrak{a}\big)_{t_2}^{\mathrm{dr}}(\bar{\mathfrak{L}}_2(\theta_2),\theta_2-\theta_1)\Big[\partial_X\bar{r}_{t_2}(\bar{\mathfrak{L}}_2(\theta_2),\theta_2)\Big]\frac{\mathrm{sgn}\big(\bar{\mathfrak{L}}_2(\theta_1)-\bar{\mathfrak{L}}_1(\theta_1)-(t_2-t_1)\theta_1\big)}{2}\\
&- \bar{r}_{t_2}(\bar{\mathfrak{L}}_2(\theta_2),\theta_2)\big(\mathfrak{a}\big)_{t_1}^{\mathrm{dr}}(\bar{\mathfrak{L}}_1(\theta_1),\theta_2-\theta_1)\Big[\partial_X\bar{r}_{t_1}(\bar{\mathfrak{L}}_1(\theta_1),\theta_1)\Big]\frac{\mathrm{sgn}\big(\bar{\mathfrak{L}}_2(\theta_2)-\bar{\mathfrak{L}}_1(\theta_2)-(t_2-t_1)\theta_2\big)}{2}\\
&+ \Big[\partial_X\bar{r}_{t_1}(\bar{\mathfrak{L}}_1(\theta_1),\theta_1)\Big]\Big[\partial_X\bar{r}_{t_2}(\bar{\mathfrak{L}}_2(\theta_2),\theta_2)\Big]\int_{-\infty}^{\infty}d\theta'\,\big(\mathfrak{a}\big)_{t_1}^{\mathrm{dr}}(\bar{\mathfrak{L}}_1(\theta_1),\theta_1-\theta')\big(\mathfrak{a}\big)_{t_2}^{\mathrm{dr}}(\bar{\mathfrak{L}}_2(\theta_2),\theta_2-\theta')\\
&\times \Big[\int_{-\infty}^{\infty}dX'\bar{r}_0(X',\theta')\frac{\mathrm{sgn}\big(\bar{\mathfrak{L}}_1(\theta')-t_1\theta'-X'\big)}{2}\frac{\mathrm{sgn}\big(\bar{\mathfrak{L}}_2(\theta')-t_2\theta'-X'\big)}{2}\Big],
\end{aligned}
$$

Here the derivative is defined as $\Big[\partial_X\bar{r}_{t_1}(\bar{\mathfrak{L}}_1(\theta_1),\theta_1)\Big] = \Big[\partial_X\bar{r}_{t_1}(X,\theta_1)\Big]_{X=\bar{\mathfrak{L}}_1(\theta_1)}$.

We can find equal time correlation by setting $\tau_1=\tau_2=t$ in Eq. (108) which gives

$$
\begin{aligned}
\mathcal{C}_{tt}(x_1,\theta_1,x_2,\theta_2) &= \bar{r}_t(\bar{\mathfrak{L}}_1(\theta_1),\theta_1)\delta(\bar{\mathfrak{L}}_1(\theta_1)-\bar{\mathfrak{L}}_2(\theta_2))\delta(\theta_1-\theta_2) \hspace{2cm}(109)\\
&+ \bar{r}_t(\bar{\mathfrak{L}}_1(\theta_1),\theta_1)\big(\mathfrak{a}\big)_\tau^{\mathrm{dr}}(\bar{\mathfrak{L}}_2(\theta_2),\theta_2-\theta_1)\Big[\partial_X\bar{r}_t(\bar{\mathfrak{L}}_2(\theta_2),\theta_2)\Big]\frac{\mathrm{sgn}\big(\bar{\mathfrak{L}}_2(\theta_1)-\bar{\mathfrak{L}}_1(\theta_1)\big)}{2}\\
&- \bar{r}_t(\bar{\mathfrak{L}}_2(\theta_2),\theta_2)\big(\mathfrak{a}\big)_\tau^{\mathrm{dr}}(\bar{\mathfrak{L}}_1(\theta_1),\theta_1-\theta_2)\Big[\partial_X\bar{r}_\tau(\bar{\mathfrak{L}}_1(\theta_1),\theta_1)\Big]\frac{\mathrm{sgn}\big(\bar{\mathfrak{L}}_2(\theta_2)-\bar{\mathfrak{L}}_1(\theta_2)\big)}{2}\\
&+ \Big[\partial_X\bar{r}_t(\bar{\mathfrak{L}}_1(\theta_1),\theta_1)\Big]\Big[\partial_X\bar{r}_t(\bar{\mathfrak{L}}_2(\theta_2),\theta_2)\Big]\\
&\times \int_{-\infty}^{\infty}d\theta'\,\big(\mathfrak{a}\big)_\tau^{\mathrm{dr}}(\bar{\mathfrak{L}}_1(\theta_1),\theta_1-\theta')\big(\mathfrak{a}\big)_\tau^{\mathrm{dr}}(\bar{\mathfrak{L}}_2(\theta_2),\theta_2-\theta')\\
&\times \Big[\int_{-\infty}^{\infty}dX'\bar{r}_0(X',\theta')\frac{\mathrm{sgn}\big(\bar{\mathfrak{L}}_1(\theta')-t\theta'-X'\big)}{2}\frac{\mathrm{sgn}\big(\bar{\mathfrak{L}}_2(\theta')-t\theta'-X'\big)}{2}\Big].
\end{aligned}
$$

For the hard rods we note that $(\mathfrak{a})_t^{\mathrm{dr}}(\bar{\mathfrak{L}}_1(\theta_1), \theta_1 - \theta_2) = -a(1)_t^{\mathrm{dr}}(x_1)$ hence we can simplify the expression as

$$\mathfrak{C}_{t,t}(x_1, \theta_1, x_2, \theta_2) = \frac{2\pi \bar{n}_t(x_1, \theta_1)}{(1)_t^{\mathrm{dr}}(x_1)} \delta(x_1 - x_2) \delta(\theta_1 - \theta_2) \tag{110}$$

$$+ a \frac{\mathrm{sgn}(x_2 - x_1)}{2} \left[ \bar{n}_t(x_2, \theta_2) \left[ \partial_x \bar{n}_t(x, \theta_1) \right]_{x=x_1} - \bar{n}_t(x_1, \theta_1) \left[ \partial_x \bar{n}_t(x, \theta_2) \right]_{x=x_2} \right]$$

$$+ \frac{a^2}{4} \left[ \partial_x \bar{n}_t(x, \theta_1) \right]_{x=x_1} \left[ \partial_x \bar{n}_t(x, \theta_2) \right]_{x=x_2} \left( 1 - \mathrm{sgn}(x_2 - x_1) \int_{x_1}^{x_2} dx' \ \bar{\rho}_t(x') \right).$$

When $x_1 \approx x_2 = x$ we get

$$\mathfrak{C}_{t,t}(x_1, \theta_1, x_2, \theta_2)|_{x_1 \approx x_2} = \frac{(2\pi)^2 \bar{\rho}_t(x, \theta_1)}{\left[ (1)_t^{\mathrm{dr}}(x) \right]^2} \delta(x_1 - x_2) \delta(\theta_1 - \theta_2) \tag{111}$$

$$+ a \frac{(2\pi)^2 \mathrm{sgn}(x_2 - x_1)}{2 \left[ (1)_t^{\mathrm{dr}}(x) \right]^2} \left( \bar{\rho}_t(x, \theta_2) \left[ \partial_x \bar{\rho}_t(x, \theta_1) \right] - \bar{\rho}_t(x, \theta_1) \left[ \partial_x \bar{\rho}_t(x, \theta_2) \right] \right)$$

$$+ \frac{a^2}{4} \left[ \partial_x \bar{n}_t(x, \theta_1) \right] \left[ \partial_x \bar{n}_t(x, \theta_2) \right].$$

The two-point correlation in Eq. (111) agrees with the result derived in Ref. [73] [see their Eq. (D.18)]. In the final stage of preparing this article we came across Ref. [76] which arrives at similar findings for the case of hards rods and uses a similar formulation in terms of the quasi particle phase space density.

## 5 Conclusions

We have derived the BMFT action for generic integrable, interacting particle models through a direct mapping to point-particle dynamics. Unlike previous approaches [24, 25, 61], our formulation is expressed in terms of the quasiparticle phase space *densities* $\rho$ rather than their charge contents. The key advantage is that the action can be written entirely with quantities that are standard in the MFT framework.

At the saddle point, the resulting expression for the full-counting statistics is remarkably compact and can be evaluated for *arbitrary* integrable particle models.

Extending the present analysis beyond purely ballistic fluctuations– to include diffusive or dispersive corrections– will grant access to sub-ballistic contributions in the higher-order cumulants. Moreover, the explicit mapping to point particles makes it straightforward to deform the theory, for example by introducing external force fields or by promoting the particles from ballistic to Brownian motion. These extensions are technically feasible and physically compelling, and we leave their exploration to future work.

## Acknowledgement

We thank Benjamin Doyon, Takato Yoshimura, Soumybrata Saha, and Anupam Kundu for valuable feedback and discussions. J.D.N., J.K., and A. U. are funded by the ERC Starting Grant 101042293 (HEPIQ) and the ANR-22-CPJ1-0021-01. T.S. acknowledges the financial support of the Department of Atomic Energy, Government of India, under Project Identification No. RTI 4002. T.S. thanks the support from the International

Research Project (IRP) titled 'Classical and quantum dynamics in out of equilibrium systems' by CNRS, France. J.D.N. and T.S. acknowledge the support of the International Centre for Theoretical Sciences (ICTS) during the program Indo-French Workshop on Classical and Quantum Dynamics in Out-of-Equilibrium Systems, where part of this work was completed (program code: ICTS/ifwcqm2024/12).

# A    Euler GHD as a microscopic equation of motion

We shall present here a fully microscopic derivation of the Euler GHD equation, which shows how the latter is the equation of motion of the microscopic density of the quasiparticles

$$\rho(x,\theta) \equiv \sum_i \delta(x - x_i)\delta(\theta - \theta_i) \quad . \tag{112}$$

Whenever their coordinates evolve according to the mapping of Eq. (5)

$$x_i = X_i - \frac{1}{2}\sum_{j \neq i} \mathfrak{a}_{ij}\mathrm{sgn}(x_i - x_j). \tag{113}$$

where we used the notation for the generic effective rod length

$$\mathfrak{a}_{ij} = \frac{2\pi\varphi(\theta_i - \theta_j)}{k'(\theta_i)}, \tag{114}$$

and where the time derivative of the free positions is given by the bare velocities $\dot{X}_i = \theta_i$ and the motion of the interacting particles by the dressed velocity $\dot{x}_i = v^{\mathrm{eff}}_{[\rho(x_i,\cdot)]}(\theta_i)$

Taking the time derivative of eq. 113 we get

$$\dot{x}_i = \dot{X}_i - \sum_{j \neq i} \mathfrak{a}_{ij}\delta(x_i - x_j)(\dot{x}_i - \dot{x}_j), \tag{115}$$

with $\dot{X}_i = v(\theta_i)$. The time derivative $\dot{x}_i$ will be instead in general, a function of all the particles' coordinates defined by Eq. (115). Notice that $v^{\mathrm{eff}}_{[\rho]}$ can have a generic $x$-dependence via $\rho$ but not directly. Taking the time derivative of the density of particles, we can get

$$\begin{aligned}
\partial_t \rho(x,\theta) &= \sum_i \partial_t(\delta(x - x_i)\delta(\theta - \theta_i)) \\
&= \sum_i \partial_{x_i}(\delta(x - x_i)\delta(\theta - \theta_i))v^{\mathrm{eff}}_{[\rho(x_i,\cdot)]}(\theta_i) \\
&= -\sum_i \partial_x(\delta(x - x_i)\delta(\theta - \theta_i))v^{\mathrm{eff}}_{[\rho(x,\cdot)]}(\theta) - \sum_i (\delta(x - x_i)\delta(\theta - \theta_i))\partial_x v^{\mathrm{eff}}_{[\rho(x,\cdot)]}(\theta) \\
&= -\partial_x\left(\sum_i (\delta(x - x_i)\delta(\theta - \theta_i))v^{\mathrm{eff}}_{[\rho(x,\cdot)]}(\theta)\right).
\end{aligned} \tag{116}$$

We therefore obtain

$$\partial_t \rho(x,\theta) + \partial_x\left(\rho(x,\theta)v^{\mathrm{eff}}_{[\rho(x,\cdot)]}(\theta)\right) = 0 \tag{117}$$

where we used the following delta function property $\partial_x \delta(x - y)f(y) = \partial_x(\delta(x - y)f(x))$. Also, we have introduced the effective velocity

$$\begin{aligned}
v^{\mathrm{eff}}_{[\rho(x_i,\cdot)]}(\theta_i) &= \dot{x}(x_i,\theta_i) = v(\theta_i) - \sum_{j \neq i} \mathfrak{a}_{ij}\delta(x_i - x_j)(\dot{x}(x_i,\theta_i) - \dot{x}(x_j,\theta_j)) \\
&= v(\theta_i) - \sum_{j \neq i} \mathfrak{a}_{ij}(\dot{x}(x_i,\theta_i) - \dot{x}(x_i,\theta_j)),
\end{aligned} \tag{118}$$

which, therefore, can be expressed in terms of the usual integral equation

$$v^{\text{eff}}_{[\rho(x_i,\cdot)]}(\theta_i) = v(\theta_i) - \int d\theta' \mathfrak{a}(\theta_i - \theta')\rho(x_i, \theta')(v^{\text{eff}}_{[\rho(x_i,\cdot)]}(\theta_i) - v^{\text{eff}}_{[\rho(x_i,\cdot)]}(\theta')). \tag{119}$$

Equations (117) and (119) are true at the microscopic level, namely, they are exact for any single realisation.

# B   Probability distributions of initial fluctuations

In this appendix, we compute the probability distribution functional for finding the density of (non-interacting) point particles. Let us consider a box of size $L$ containing $N$ particles in equilibrium. The system is divided into $R$ spatial cells of size $\ell$, indexed by $i = 1, 2, 3....R$, with the $i$th cell containing $r_i\ell$ particles where $r_i$ is the number density in of the cell. There are several possible density profiled $\{r_i\}$ subjected to the global constraint $\ell\sum_{i=1}^{R} r_i = N$. For large $\ell$, assuming statistical independence of each box, the probability of a density profile $\{r_i\}$ is [77]

$$\mathcal{P}\left(\{r_i\}\right) \simeq \frac{\prod_{i=1}^{R} Z_\ell(r_i\ell)}{Z_L(N)}\delta_{N,\ell\sum_{i=1}^{R} r_i} = \exp\left(-\ell\sum_{i=1}^{R}[f(r_i) - f(\bar{r})]\right)\delta_{N,\ell\sum_{i=1}^{R} r_i}, \tag{120}$$

where $Z_L(N)$ is the canonical partition function, $\delta_{a,b}$ is the Kronecker delta, $\bar{r} = N/L$ is the bulk density, and

$$f(r) = -\frac{1}{\ell}\log\left[Z_\ell(r\ell)\right] \tag{121}$$

is the free energy density.

In the grand canonical ensemble, the Kronecker delta is replaced with the fugacity, and we find that the distribution is given by

$$\mathcal{P}\left(\{r_i\}\right) \asymp \exp\left(-\ell\sum_{i=1}^{R}\left[f(r_i) - f(\bar{r}) - (r_i - \bar{r})f'(\bar{r})\right]\right), \tag{122}$$

where $f'(\bar{r})$ is the chemical potential. For inhomogeneous systems with typical density profile $\bar{r}_i$, this generalises to

$$\mathcal{P}\left(\{r_i\}\right) \asymp \exp\left(-\ell\sum_{i=1}^{R}\left[f(r_i) - f(\bar{r}_i) - (r_i - \bar{r}_i)f'(\bar{r}_i)\right]\right). \tag{123}$$

Taking the continuum limit $\sum_i \ell \to \int dX$, we get

$$\mathcal{P}\left[r(X)\right] \asymp \exp\left(-\int dX\left[f(r) - f(\bar{r}) - (r - \bar{r})f'(\bar{r})\right]\right) \tag{124}$$

for fluctuations of the density profile $r(X)$ around the average profile $\bar{r}(X)$ subject to constraint $\int dX\, r(X) = N$.

For our work the relevant quantity is the probability distribution of the phase space density $r(X, \theta)$. For this the above argument can be straightforwardly extended using

independence of $(X, \theta)$ coordinates for point particles, which gives the probability of a phase space density fluctuation $r(X, \theta)$,

$$\mathcal{P}\left[r(X, \theta)\right] \asymp \exp\left(-\tilde{\mathcal{F}}[r(x, \theta)]\right), \tag{125}$$

$$\tilde{\mathcal{F}}[r] = \int d\theta \int dX \left[f(r) - f(\bar{r}) - (r - \bar{r})f'(\bar{r})\right]. \tag{126}$$

Here, $\tilde{\mathcal{F}}[r]$ is the free energy cost of finding a profile $r(X, \theta)$ different from the average profile $\bar{r}(X, \theta)$. The free energy density $f(r(X, \theta))$ depends on the particle statistics and is obtained from the partition function

$$f(r) = \begin{cases} r \log(r) - r & \text{for classical particles} \\ r \log r + \eta(1 - \eta r) \log(1 - \eta r) & \text{for Quantum systems} \end{cases}, \tag{127}$$

with $\eta = \pm 1$ for Fermions and Bosons respectively. Substituting the Eq. (127) in Eq. (126) we obtain the free energy cost as

$$\tilde{\mathcal{F}}[r] = \int_{-\infty}^{\infty} dX \int_{-\infty}^{\infty} d\theta G(r, \bar{r}) \quad \text{with} \tag{128}$$

$$G(r, \bar{r}) = \begin{cases} r \log\left(\frac{r}{\bar{r}}\right) - (r - \bar{r}) & \text{for Classical particles,} \\ r \log\left(\frac{r}{\bar{r}}\right) + (1 - r) \log\left(\frac{1-r}{1-\bar{r}}\right) & \text{for Fermions,} \\ r \log\left(\frac{r}{\bar{r}}\right) - (1 + r) \log\left(\frac{1+r}{1+\bar{r}}\right) & \text{for Bosons.} \end{cases}$$

In the next section we derive these results from first principles based on combinatorial arguments for the classical free particles.

## B.1 Probability of the initial profile: classical free particles

The phase space density $r(X, \theta)$ represents the number of particles, $r(X, \theta)\Delta X \Delta \theta$, within a region of size $\Delta X \Delta \theta$ centred at $(X, \theta)$ on the phase space. For computing the probability (125), we partition the phase space (see Fig. 3) along $X$-axis into $R$ strips of width $\Delta X$ containing $\{n_i\}_{i=1}^{R}$ particles with $n_i = \Delta X \int d\theta\, r(X_i, \theta)$. Within each strip, the particles are further distributed into $M$ sub-partitions based on their momentum, $\{n_{i,j}\}_{j=1}^{M} \ \forall \ i = 1 \cdots M$ with $n_{i,j} = \Delta X \Delta \theta\, r(X_i, \theta_j)$ being the number of particles in the box $\Delta X \Delta \theta$ centred at $(X_i, \theta_j)$ on the phase space.

For independently distributed particles (or point particles), the probability of a configuration $\{n_i\}$ across all strips is the product of individual probabilities expressed as

$$\mathcal{P}(\{n_i\}_{i=1}^{R}) = \prod_{i=1}^{R} P(\{n_{i,j}\}_{j=1}^{M}), \tag{129}$$

where $P(\{n_{i,j}\}_{j=1}^{M})$ is the probability of observing the occupation $\{n_{i,j}\}_{j=1}^{M}$, which typically contains $\{\bar{n}_{i,j}\}_{j=1}^{M}$ with $\bar{n}_{i,j} = \Delta X d\theta \bar{r}(X_i, \theta)$. This probability can be obtained using the conditional probability as follows:

$$P(\{n_{i,j}\}_j) = P\left(\{n_{i,j}\}_j \middle| n_i = \sum_j n_{i,j}\right) P(n_i), \tag{130}$$

where $P(\{n_{i,j}\}_j | n_i)$ is the conditional probability of finding $\{n_{i,j}\}_j$ number of particles in the $M$ boxes of the strip and $P(n_i)$ is the probability of finding $n_i$ particles in the $i^{\text{th}}$ strip. The conditional probability follows the multinomial distribution, since the particles are

distributed independently in the momentum space with probability $p_{i,j} = \bar{n}_{i,j}/\bar{n}_i$ where $\bar{n}_i = \sum_j \bar{n}_{i,j}$. Expressing the multinomial in the large deviation form simplifies to

$$P(\{n_{i,j}\}_j | n_i) \asymp \exp\left( -\sum_{j=1}^{M} n_{i,j} \log\left( \frac{n_{i,j}}{n_i} \frac{\bar{n}_i}{\bar{n}_{i,j}} \right) \right). \tag{131}$$

On the other hand, the probability of finding $n_i$ particles in the $i^{\text{th}}$ strip follows a binomial distribution. We can express it in the large deviation form by using $n_i \ll N$ and $\Delta X \ll L$ to get

$$P(n_i) \asymp \exp\left( -n_i \log\left( \frac{n_i}{N} \right) + n_i \log\left( \frac{\Delta X}{L} \right) + n_i - \Delta X \frac{N}{L} \right). \tag{132}$$

Substituting Eq. (131) and Eq. (132) in Eq. (130) we find

$$P(\{n_{i,j}\}_j = \{r(X_i,\theta_j)\Delta X\Delta\theta\}_j) \asymp$$
$$\exp\left( -\Delta X \left( \int d\theta r(X_i,\theta) \log\left( \frac{r(X_i,\theta)}{\bar{r}(X_i,\theta)} \right) - \int d\theta (r(X_i,\theta) - \bar{r}(X_i,\theta)) \right) \right), \tag{133}$$

where $n_{i,j} = \Delta X\Delta\theta\, r(X_i,\theta_j)$ and $\bar{n}_{i,j} = \Delta X\Delta\theta\, \bar{r}(X_i,\theta_j)$. Finally, substituting Eq. (133) in Eq. (129) yields the

$$\mathcal{P}[r] \asymp \exp\left( -\int d\theta \int dX \left[ r \log\left( \frac{r}{\bar{r}} \right) - (r - \bar{r}) \right] \right), \tag{134}$$

recovering the result for the classical particles in Eq. (128). A similar approach is used in the next section to compute the probability distribution for the phase space density of the hard rods.

## B.2 Probability distribution of initial fluctuations for hard rods

We follow similar arguments as in the previous sections for hard rods of size $a$. Namely, we compute the probability distribution function of observing a phase space density $\rho(x,\theta)$ relative to a typical profile $\bar{\rho}(x,\theta)$, by adapting the combinatorial approach used for point particles described in Appendix B.1. The main difference from the point particles case is the finite size of the rods, which introduces correlations. As a result, the rods inside the strip and outside are no longer independent. Nevertheless we handle this correlation by mapping the hard rods to point particles using a coordinate transformation

$$X_i = x_i - \frac{N-i}{2}a, \tag{135}$$

where $x_i$ denotes the position of the rods and $X_i$ the mapped point particles. As a consequence of mapping, a strip of width $\Delta x$ containing $n_i = dx \int d\theta \rho(x_i,\theta)$ rods now corresponds to an effective width $\Delta X = \Delta x - n_i a$, while the system size contracts to $L \to L - Na$. Hence, the probability $P(n_i)$ of finding $n_i$ rods in the $i^{\text{th}}$ strip follows a binomial distribution with contracted widths $i.e.$ $\Delta x \to \Delta X = \Delta x - n_i a$ and $L \to L - Na$, which gives

$$P(n_i) = \frac{N!}{(n_i)!(N-n_i)!} \left( \frac{\Delta x - n_i a}{L - Na} \right)^{n_i} \left( 1 - \frac{\Delta x - n_i a}{L - Na} \right)^{N-n_i}. \tag{136}$$

In the large deviation form, it can be expressed as

$$P(n_i) \asymp \exp\left(-n_i \log\left(\frac{n_i}{N}\right) + n_i \log\left(\frac{\Delta x - n_i a}{L - Na}\right) + \frac{n_i - \Delta x \frac{N}{L}}{1 - a\frac{N}{L}}\right), \qquad (137)$$

where we used the approximation that $n_i \ll N$ and $\Delta x \ll L$. Substituting the multinomial distribution in Eq. (131) and Eq. (137) in Eq. (130) we find

$$P(\{n_{i,j}\}_j = \{\rho(x_i, \theta_j)\Delta x \Delta\theta\}_j) \qquad (138)$$

$$\asymp \exp\left(-\Delta x\left(\int d\theta \rho(x_i, \theta) \log\left(\frac{\rho(x_i, \theta)}{1 - a\rho(x_i)}\frac{1 - a\bar\rho(x_i)}{\bar\rho(x_i, \theta)}\right)\right) + \Delta x\left(\frac{\rho(x_i) - \bar\rho(x_i)}{1 - a\bar\rho(x_i)}\right)\right).$$

Here, we used the relations

$$(a) \;\; n_i = \rho(x_i)\Delta x, \qquad (b) \;\; \bar n_i = \bar\rho(x_i)\Delta x, \qquad (c) \;\; n_{i,j} = \rho(x_i, \theta_j)\Delta x \Delta\theta, \qquad (139)$$
$$(d) \;\; \bar n_{i,j} = \bar\rho(x_i, \theta_j)\Delta x \Delta\theta, \qquad (e) \;\; N = \bar\rho L, \qquad\qquad\qquad\qquad\quad (140)$$

where $\bar\rho(x, \theta)$ and $\bar\rho(x) = \int d\theta \bar\rho(x, \theta)$ are the typical phase-space density and number density, respectively. We express Eq. (138) as a function of the point particle coordinates by using the relation Eq. (17), which for hard rods is given by

$$r(X, \theta) = \frac{\rho(x, \theta)}{1 - a\rho(x)} \text{ and } \bar r(X, \theta) = \frac{\bar\rho(x, \theta)}{1 - a\bar\rho(x)}. \qquad (141)$$

Using the relation Eq. (141) in Eq. (138) we recast the distribution in the point particle density $r(X, \theta)$ as

$$P(\{n_{i,j}\}_j = \{\rho(x_i, \theta_j)\Delta x \Delta\theta\}_j)$$

$$\asymp \exp\left(-\Delta X \int d\theta\left[r(X_i, \theta) \log\left(\frac{r(X_i, \theta)}{\bar r(X_i, \theta)}\right) - \left(r(X_i) - \bar r(X_i)\right)\right]\right), \qquad (142)$$

where $\Delta X = \Delta x/(1 - a\rho(x))$. Finally, substituting Eq. (142) in Eq. (129) we obtain

$$\mathcal{P}[\rho] \asymp \exp\left(-\int d\theta \int dX\left[r \log\left(\frac{r}{\bar r}\right) - (r - \bar r)\right]\right), \qquad (143)$$

where $r(X, \theta)$ and $\bar r(X, \theta)$ are obtained from Eq. (141). Strikingly, this matches with the case of free particle Eq. (134), with $r(X, \theta)$ and $\bar r(X, \theta)$ now representing the mapped observed and typical densities.

This suggests a general principle: for interacting systems admitting a free-particle mapping via transformations like Eq. (5), their large deviation functionals can be expressed as Eq. (128), provided the density is the mapped to point particle density obtained from Eq. (17).

## C  Microscopic calculation for point particles

Integrated mass current through the origin during time $T$ is defined by

$$\tilde Q_T = R_T - R'_T, \qquad (144)$$

where $R_T$ is the number of particles which start at any position $\leq 0$ and reach a position $> 0$ at time $T$. Similarly, $R'_T$ is the number of particles which start at any position $> 0$

and reach a position $\leq 0$ at time $T$. The cumulant generating function of $\tilde{Q}_T$ is defined by

$$\mu(\lambda) = \ln \langle e^{\lambda \tilde{Q}_T} \rangle \tag{145}$$

where the angular brackets denote the average over the initial position and velocity distribution of the particles.

We shall consider the case of hard point particles with equal mass ($m = 1$). The hard-core interaction is imposed by the non-crossing condition of the trajectories. Particles follow straight line trajectories between collisions, and in each binary collision, particles exchange their momentum.

An important realisation is that for each history of hard-core point particles, there is an associated history of non-interacting point particles, obtained by swapping the particle identity in each collision event. This implies that the distribution of particle positions at time $T$, independent of their identity, is the same for interacting and non-interacting particles. The current $\tilde{Q}_T$ is independent of the particle's identity and only depends on the distribution of their position at time $T$. Consequently, the cumulant generating function Eq. (145) for hard point rods is the same as for non-interacting point particles, which is straightforward to determine.

For the non-interacting case, we use the independence of the particles to write

$$\left\langle e^{\lambda Q_T} \right\rangle = \left\langle e^{\lambda R_T} \right\rangle \left\langle e^{-\lambda R'_T} \right\rangle = \left\langle \prod_i \overline{F}_\lambda(X_i(0), T) \right\rangle_{\mathbf{X}(0)} \tag{146}$$

where index $i$ denotes the index of particles, located initially at position $X_i(0)$. The angular bracket $\langle \rangle_{\mathbf{X}(0)}$ denotes average over initial particle arrangements. The average over initial velocity of particles is contained in the single-particle function

$$\overline{F}_\lambda(Y, T) = \begin{cases} 1 + (e^\lambda - 1) \overline{\Theta(X(T))}|_{X(0)=Y} & \text{for } Y \leq 0, \\ 1 + (e^{-\lambda} - 1) \overline{\Theta(-X(T))}|_{X(0)=Y} & \text{for } Y > 0, \end{cases} \tag{147}$$

with $\Theta(X)$ being the Heaviside theta function and the overline denoting average over initial velocity of the particle initially at position $X(0) = Y$.

The velocity average $\overline{\theta(X(T))}|_{X(0)=Y}$ is simple to evaluate using the probability for a single point particle to be at position $X$ at time $T$, starting at $Y$, irrespective of its velocity

$$g_T(X|Y) = \int d\theta \, p(\theta) \delta(X - Y - \theta T) = \left(\frac{2\pi}{\beta}\right)^{-1/2} \exp\left(-\frac{\beta}{2T^2}(X - Y)^2\right) \tag{148}$$

where initial velocity distribution is chosen to be Maxwellian $p(\theta) = \left(\frac{2\pi}{\beta}\right)^{-1/2} \exp\left(-\frac{\beta}{2}\theta^2\right)$ with inverse temperature $\beta$.

We find that $\overline{F}_\lambda(Y, T)$ has a scaling form

$$\overline{F}_\lambda(Y, T) = \begin{cases} \mathfrak{f}_\lambda\left(-Y\sqrt{\frac{\beta}{2T^2}}\right) & \text{for } Y \leq 0, \\ \mathfrak{f}_{-\lambda}\left(Y\sqrt{\frac{\beta}{2T^2}}\right) & \text{for } Y > 0, \end{cases} \tag{149}$$

with

$$\mathfrak{f}_\lambda(\eta) = 1 + \left(e^\lambda - 1\right)\frac{1}{2}\text{Erfc}(\eta) \tag{150}$$

where the $\mathrm{Erfc}(X)$ is the complimentary error function.

For the average over initial particle positions in (146), we use that initially, the particles are uniformly distributed in position with average density profile $r(X)$. This leads to the generating function

$$\left\langle e^{\lambda \tilde{Q}_T} \right\rangle = \prod_X \left\{ (1 - r(X)dX) + r(X)dX \, \overline{F}_\lambda(X, T) \right\} \tag{151}$$

which yields the cumulant generating function Eq. (145):

$$\mu(\lambda) = \int_{-\infty}^{\infty} dX \, r(X) \left( \overline{F}_\lambda(X, T) - 1 \right). \tag{152}$$

Substituting Eqs. (149) and (150) we arrive at an explicit expression for the cumulant generating function.

$$\mu(\lambda) = T \sqrt{\frac{2}{\beta}} \int_0^{\infty} d\eta \, \left\{ \tilde{r}(-\eta) \left( \mathfrak{f}_\lambda(\eta) - 1 \right) + \tilde{r}(\eta) \left( \mathfrak{f}_{-\lambda}(\eta) - 1 \right) \right\} \tag{153}$$

where we defined $r(X) = \tilde{r}\left( X \sqrt{\frac{\beta}{2T^2}} \right)$. This expression simplifies for the domain wall initial density profile $r(x) = r_- \theta(-x) + r_+ \theta(x)$, leading to an explicit expression

$$\mu(\lambda) = \frac{T}{\sqrt{2\pi\beta}} \left( r_-(e^\lambda - 1) + r_+(e^{-\lambda} - 1) \right) \tag{154}$$

which is reproduced using the BMFT in (46) by setting $\bar{q}_0(X, \theta) = r(X)p(\theta)$ with the Maxwellian $p(\theta)$ given in Eq. (148).

# D  Two-point normal modes correlations

In this appendix, we provide the details of computing the 2-point normal mode correlations. We compute the generating function given in Eq. (104) using the path integral formulation described in section 2 as

$$\exp\left( T\mathcal{G}(\lambda_1, \lambda_2) \right) = \int \mathcal{D}[r_t(X, \theta)] \int \mathcal{D}[\hat{r}_t(X, \theta)] \exp\left( \tilde{\mathcal{S}}[r_t(X, \theta), \hat{r}_t(X, \theta)] \right), \quad \text{with} \tag{155}$$

$$\tilde{\mathcal{S}}[r_t(X, \theta), \hat{r}_t(X, \theta)] = -\tilde{\mathcal{F}}[r_0(X, \theta)] + T\lambda_1 r_{t_1}\left( \mathfrak{L}_1(\theta_1), \theta_1 \right) + T\lambda_2 r_{t_2}\left( \mathfrak{L}_2(\theta_2), \theta_2 \right) \tag{156}$$

$$- \int_{-\infty}^{\infty} d\theta \int_{-\infty}^{\infty} dX \int_0^T dt \, \hat{r}_t(X, \theta) \Big[ \partial_t r_t(X, \theta) + u(\theta)\partial_y r_t(X, \theta) \Big],$$

where $\hat{r}_t(X, \theta)$ is the auxiliary field that enforces the GHD for free particles [Eq. (25)] as a constraint. To compute the generating function at large $T$, we apply Euler scaling by defining

$$Z = \frac{X}{T}, \quad \tau = \frac{t}{T}, \quad q_\tau(Z, \theta) = r_t(X, \theta), \quad p_\tau(Z, \theta) = \hat{r}_t(X, \theta). \tag{157}$$

The action in Eq. (156) rescales as

$$\tilde{\mathcal{S}}[r_t(X, \theta), \hat{r}_t(X, \theta)] = T\mathcal{S}[q_\tau(Z, \theta), p_\tau(Z, \theta)] \tag{158}$$

$$\mathcal{S}[q_\tau(Z, \theta), p_\tau(Z, \theta)] = -\mathcal{F}[q_0(Z, \theta)] + \lambda_1 q_{\tau_1}\left( \mathfrak{l}_1(\theta_1), \theta_1 \right) + \lambda_2 q_{\tau_2}\left( \mathfrak{l}_2(\theta_2), \theta_2 \right)$$

$$- \int_{-\infty}^{\infty} d\theta \int_{-\infty}^{\infty} dZ \int_0^1 d\tau \, p_\tau(Z, \theta) \Big[ \partial_\tau q_\tau(Z, \theta) + \theta \partial_z q_\tau(Z, \theta) \Big], \tag{159}$$

where $\mathcal{F}[q_0] = \tilde{\mathcal{F}}[r_0]/T$ and the scaled coordinates $\mathfrak{l}_{1/2}(\theta) = \mathfrak{L}_{1/2}(\theta)/T$ [ Eqs. (102) and (103)] are

$$\mathfrak{l}_1(\theta) = \frac{x_1}{T} + \frac{1}{2} \int_{-\infty}^{\infty} d\theta' \int_{-\infty}^{\infty} dZ' \ \mathfrak{a}(\theta - \theta') \mathrm{sgn}\big(\mathfrak{l}_1(\theta') - Z'\big) q_{\tau_1}(Z', \theta'), \tag{160}$$

$$\mathfrak{l}_2(\theta) = \frac{x_2}{T} + \frac{1}{2} \int_{-\infty}^{\infty} d\theta' \int_{-\infty}^{\infty} dZ' \ \mathfrak{a}(\theta - \theta') \mathrm{sgn}\big(\mathfrak{l}_2(\theta') - Z'\big) q_{\tau_2}(Z', \theta'). \tag{161}$$

For large $T$, the path integral in Eq. (155) is dominated by the saddle point configuration $p_0^*(Z, \theta)$ and $q_0^*(Z, \theta)$ and they satisfy the following variational saddle point equations

$$\frac{\delta \mathcal{S}}{\delta q_0(Z, \theta)} = p_0^*(Z, \theta) - \frac{\delta \mathcal{F}[q_0^*(Z, \theta)]}{\delta q_0(Z, \theta)} = 0, \tag{162a}$$

$$\frac{\delta \mathcal{S}}{\delta q_1(Z, \theta)} = p_1^*(Z, \theta) = 0, \tag{162b}$$

$$\frac{\delta \mathcal{S}}{\delta p_\tau(Z, \theta)} = \partial_\tau q_\tau^*(Z, \theta) + \theta \partial_z \ q_\tau^*(Z, \theta) = 0, \tag{162c}$$

$$\frac{\delta \mathcal{S}}{\delta q_\tau(Z, \theta)} = \partial_\tau p_\tau^*(Z, \theta) + \theta \partial_z \ p_\tau^*(Z, \theta) = -h_\tau(Z, \theta). \tag{162d}$$

Here, the source term in Eq. (162d) for the auxiliary fields is given by

$$h_\tau(Z, \theta) = \sum_{i=1}^{2} h_\tau^{(i)}(Z, \theta) \ \ \text{where} \tag{163}$$

$$h_\tau^{(i)}(Z, \theta) = \lambda_i \delta(\tau - \tau_i) \delta(Z - \mathfrak{l}_i^*(\theta_i)) \delta(\theta - \theta_i) \tag{164}$$

$$+ \lambda_i \delta(\tau - \tau_i) \big(\mathfrak{a}\big)_{\tau_i}^{\mathrm{dr}}(\mathfrak{l}_i^*(\theta_i), \theta_i - \theta) \frac{\mathrm{sgn}\big(\mathfrak{l}_i^*(\theta) - Z\big)}{2} [\partial_{Z'} q_{\tau_i}^*\big(Z', \theta_i\big)]_{Z'=\mathfrak{l}_i^*(\theta_i)}.$$

Here, the first term on the right-hand side is due to the variation of the density at the location $(\mathfrak{l}_i^*(\theta_i), \theta_i)$ while the second term is due to the variation of $\mathfrak{l}_i^*(\theta_i)$ for both $i = 1, 2$. In Eq. (164), the dressing operation is, $(h)_{\tau_i}^{\mathrm{dr}}\big(\mathfrak{l}_i^*(\theta), \theta\big) \equiv (h)^{\mathrm{dr}}[q_{\tau_i}^*]\big(\mathfrak{l}_i^*(\theta), \theta\big)$, defined as

$$(h)_{\tau_i}^{\mathrm{dr}}\big(\mathfrak{l}_i^*(\theta), \theta\big) = h(\theta) + 2\pi \int_{-\infty}^{\infty} d\theta' \varphi(\theta - \theta') q_{\tau_i}\big(\mathfrak{l}_i^*(\theta'), \theta'\big) (h)_{\tau_i}^{\mathrm{dr}}\big(\mathfrak{l}_i^*(\theta'), \theta'\big). \tag{165}$$

Note that this matches the dressing in the interacting coordinates

$$(h)^{\mathrm{dr}}[q_{\tau_i}^*]\big(\mathfrak{l}_i^*(\theta), \theta\big) = (h)^{\mathrm{dr}}[n_{t_1}^*](x_1, \theta) \tag{166}$$

since $n_{t_i}^*(x_i, \theta) = q_{\tau_i}^*(\mathfrak{l}_i^*(\theta), \theta)$. Using Eq. (162a) and the scaled free energy cost given in Eq. (40), we can find the initial condition for the phase space density $q_0^*(Z, \theta)$ by solving

$$\frac{q_0^*(Z, \theta)}{1 - \eta q_0^*(Z, \theta)} = \frac{\bar{q}_0(Z, \theta)}{1 - \eta \bar{q}_0(Z, \theta)} \exp(p_0^*(Z, \theta)). \tag{167}$$

To obtain $p_0^*(Z, \theta)$, we solve Eq. (162d) using the method of characteristics (basically travelling along the field), which gives

$$\frac{d}{ds} p_{\tau+s}^*\big(Z + s\theta, \theta\big) = -h_{\tau+s}(z + s\theta, \theta), \tag{168}$$

Integrating from $s = -\tau$ to 0 and applying the boundary condition $p_1^*(Z, \theta) = 0$ [Eq. (162b)], yields

$$p_0^*(Z, \theta) = \int_0^1 dr \ h_r(Z + r\theta, \theta). \tag{169}$$

Substituting the expression of the source term [Eq. (163)] in Eq. (169) we get

$$
\begin{aligned}
p_0^*(Z,\theta) = {} & \lambda_1 \delta(Z + \tau_1 \theta - \mathfrak{l}_1^*(\theta_1))\delta(\theta - \theta_1) \tag{170}\\
& + \lambda_1 (\mathfrak{a})_{\tau_1}^{\mathrm{dr}}(\mathfrak{l}_1^*(\theta_1),\theta_1 - \theta)\frac{\mathrm{sgn}\big(\mathfrak{l}_1^*(\theta) - \tau_1 \theta - Z\big)}{2}[\partial_{Z'} q_{\tau_1}^*(Z',\theta_1)]_{Z' = \mathfrak{l}_1^*(\theta_1)} \\
& + \lambda_2 \delta(Z + \tau_2 \theta - \mathfrak{l}_2^*(\theta))\delta(\theta - \theta_2) \\
& + \lambda_2 (\mathfrak{a})_{\tau_2}^{\mathrm{dr}}(\mathfrak{l}_2^*(\theta_2),\theta_2 - \theta)\frac{\mathrm{sgn}\big(\mathfrak{l}_2^*(\theta) - \tau_2 \theta - Z\big)}{2}[\partial_{Z'} q_{\tau_2}^*(Z',\theta_2)]_{Z' = \mathfrak{l}_2^*(\theta_2)}.
\end{aligned}
$$

Using the saddle point solutions computed in Eqs. (167) and (170) in Eq. (155) we get the generating function given in Eq. (105).

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
