# Peer review of "Ballistic macroscopic fluctuation theory via mapping to point particles"

_SciPost Physics_

## Round 1 · Referee Report · Anonymous (Referee 1) · 2025-10-24

Report

The authors use a generalized hard rod mapping to express asymptotic quasi-particle trajectories in integrable systems to those of non-interacting particles. These facilitate a straightforward computation of large fluctuations and correlations after reversing the mapping from non-interacting to interacting particles.

Some comments and questions:

  1. It is not clear whether the central equations 5 and 9 of the generalized hard rod transformation are simply postulated or whether they refer to previous work (specifically ref. 69, 70 being a detailed investigation of a particular example).

  2. The paper is at times difficult to follow due to extensive derivations and equations in the main text, e.g. BMFT of point particles in section 2, and two-point correlations on section 5, the majority of which could be moved to the appendix while preserving the main message of the paper.

  3. What is a "typical equilibrium profile" mentioned in 26? I could not find a statement as to how it is characterized. This also brings me to a conceptual problem - In eq. 21 the large deviations function is expressed as an average over a "typical" phase space density while large deviations invariable result from "atypical" initial configurations. How is this reconciled?

  4. Section 3: "... we have assumed that the statistical nature of the initial profile, when described in the point particle density ..., is governed by the probability density functional with the large deviation function..."

What precisely does this assumption encompass?

  1. In Figure 4 the deviations from Gaussianity are appreciable only for the smallest time (T=10). Presumably this is due to direct sampling of large values becoming exponentially hard with time. On the other hand, it is not clear from the presented data what part of this is due to finite-time effects. In principle large deviations can be sampled efficiently by tilting the measure.

  2. The identification of cumulants and correlators with derivatives of the asymptotic form of the their respective generating functions in eqs. 82, 95 123 requires additional regularity assumptions, see e.g. Ž Krajnik, J Schmidt, V Pasquier, E Ilievski, T Prosen Physical Review Letters 128 (16), 160601. This is more than a technical detail as the regularity condition is known to be violated in a number of integrable models.

  3. In the Conclusions, the authors claim that the method can be applied to "generic" and "arbitrary" integrable models. At the moment this does not seem to be the case for classical integrable models. For example eq. 29 gives the free energy of classical particles which is correct but incomplete whereas 142 refers to classical systems which is wrong.

Specifically, it is known that description of certain classical integrable models involve radiative modes nor can solitons always be treated as classical particles, see e.g.:

A Bastianello, B Doyon, GMT Watts, T Yoshimura SciPost Physics 4 (6), 045 R Koch, JS Caux, A Bastianello Journal of Physics A: Mathematical and Theoretical 55 (13), 134001 A Bastianello, Ž Krajnik, E Ilievski Physical Review Letters 133 (10), 107102

To the best of my knowledge, a derivation of generalized hydrodynamics from kinetics of a generic classical integrable systems remains an open problem precisely because of this difficulty.

Minor comments/typos:

page 3: hard rods gas -> hard rod gas page 5: in the express ... in the normal mode density (?) eqs 1, 14 and others: the order of integrals over x and theta changes sporadically Appendices: Hard rods -> hard rods In 1.2 after eq. 14 the free coordinates (X, k) are mentioned, but it is unclear what k is from the surrounding explanation.

Overall, the work explores a natural idea to quantify dynamics of quasi-particles within a hydrodynamic framework and uses it to recover some previously obtained results. However, as noted above, some points require further clarification and it is not clear how different the method is from related hydrodynamic techniques used in the field.

Recommendation

Ask for major revision

---

## Round 1 · Referee Report · Anonymous (Referee 2) · 2025-12-31

Report

The authors derive Ballistic Macroscopic Fluctuation Theory for a class of integrable models by means of a mapping to point particles. They re-derive known results for full counting statistics and long-range correlation functions.

This is a rather technical but interesting piece of work and certainly should be published. However, given that the work appears to be largely concerned with the rederivation of known results, it is not immediately obvious that it fulfils the criteria for SciPost Physics.

I have a number of comments and questions I would ask the authors to address before the paper can be recommended for publication in SciPost Physics.

  1. The paper is very technical and I don't think it will be clear to most readers what the advantage of the new approach of deriving BMFT is compared to Refs [24-26]. I think the authors should explain in more detail what can be done in the new approach which was not possible in the one of Refs [24-26]. The relevant SciPost Physics criterion is "Open a new pathway in an existing or a new research direction, with clear potential for multi-pronged follow-up work." How does this work satisfy this criterion?

  2. What is the physical importance of the normal-mode two-point function determined in Section 4 (given that the normal mode density is very complicated in terms of the microscopic degrees of freedom)? Can it be measured in experiments? If so the authors should explain. What is its theoretical importance? How difficult is it to evaluate in practice, and what can be learned from it?

  3. Throughout the paper the authors stress that their approach applies to general integrable models. However, they only consider a theory with a single particle species and diagonal scattering. If the generalization to models with several particle species and scattering matrices as obvious as the authors' comments suggest, they should consider giving the relevant equations in an Appendix.

  4. Regarding the observed non-Gaussianity in Fig 4: the saddle-point approximation holds only in the large-T limit, for which the data in Fig.4 is well described by a Gaussian. How can the authors be sure that the corrections to the saddle-point approximation for T=10 are negligible? I assume that the authors would claim that the numerical results show this, but the quality of the numerical data collapse for the values of $Q_1$ where there is data for a range of T's is quite difficult to judge without an estimate on the error bars.

  5. What is the precise status of eqns (4) and (13)? Have they been proven (in a mathematical physics sense) or are they conjectures supported by convincing evidence? If they haven't been proven I think readers would benefit from a clear statement about what kinds of evidence supports them.

Recommendation

Ask for minor revision

---

## Editorial Decision

awaiting_resubmission